# CCDC113 stabilizes sperm axoneme and head-tail coupling apparatus to ensure male fertility

**Bingbing Wu[1,2,3†], Chenghong Long[1†], Yuzhuo Yang[4,5], Zhe Zhang[4,5], Shuang Ma[1,2,3], Yanjie Ma[1,2,3], Huafang Wei[1], Jinghe Li[1], Hui Jiang[4,5], Wei Li[1,2,3*], Chao Liu[1,2,3*]**

[1]Guangzhou Women and Children's Medical Center, Guangzhou Medical University, Guangzhou, China; [2]State Key Laboratory of Stem Cell and Reproductive Biology, Institute of Zoology, Stem Cell and Regenerative Medicine Innovation Institute, Chinese Academy of Sciences, Beijing, China; [3]University of Chinese Academy of Sciences, Beijing, China; [4]Department of Urology, Department of Reproductive Medicine Center, Peking University Third Hospital, Beijing, China; [5]Department of Urology, Peking University First Hospital Institute of Urology, Peking University, Beijing, China

**\*For correspondence:**
leways@lzu.edu.cn (WL);
liuchsdu@163.com (CL)

†These authors contributed equally to this work

**Competing interest:** The authors declare that no competing interests exist.

## eLife Assessment

This study presents an **important** finding on sperm flagellum and HTCA stabilization. The evidence supporting the authors' claims is **convincing**. The work will be of broad interest to cell and reproductive biologists working on cilium and sperm biology.

**Abstract** The structural integrity of the sperm is crucial for male fertility, defects in sperm head-tail linkage and flagellar axoneme are associated with acephalic spermatozoa syndrome (ASS) and the multiple morphological abnormalities of the sperm flagella (MMAF). Notably, impaired head-tail coupling apparatus (HTCA) often accompanies defects in the flagellum structure, however, the molecular mechanisms underlying this phenomenon remain elusive. Here, we identified an evolutionarily conserved coiled-coil domain-containing (CCDC) protein, CCDC113, and found the disruption of CCDC113 produced spermatozoa with disorganized sperm flagella and HTCA, which caused male infertility. Further analysis revealed that CCDC113 could bind to CFAP57 and CFAP91, and function as an adaptor protein for the connection of radial spokes, nexin-dynein regulatory complex (N-DRC), and doublet microtubules (DMTs) in the sperm axoneme. Moreover, CCDC113 was identified as a structural component of HTCA, collaborating with SUN5 and CENTLEIN to connect sperm head to tail during spermiogenesis. Together, our studies reveal that CCDC113 serve as a critical hub for sperm axoneme and HTCA stabilization in mice, providing insights into the potential pathogenesis of infertility associated with human *CCDC113* mutations.

## Introduction

Male fertility relies on the continuous production of spermatozoa through a complex developmental process known as spermatogenesis. Spermatogenesis involves three primary stages: spermatogonia mitosis, spermatocyte meiosis, and spermiogenesis. During spermiogenesis, spermatids undergo complex differentiation processes to develop into spermatozoa, which includes nuclear elongation, chromatin remodeling, acrosome formation, cytoplasm elimination, and flagellum development

(*Hermo et al., 2010*). The integrity of spermatozoa is essential for their migration through the female reproductive tract and subsequent successful fertilization. An intact spermatozoon contains properly formed sperm head, head-tail coupling apparatus (HTCA), and flagellum (*Parker, 2020*; *Roldan, 2019*). Numerous components have been identified in the sperm HTCA and flagellum that are essential for the sperm integrity (*Lehti and Sironen, 2017*; *Wu et al., 2020*). Defects in sperm flagellum and HTCA can lead to reduced sperm motility or abnormal sperm morphology, termed as multiple morphological abnormalities of the sperm flagella (MMAF) or acephalic spermatozoa syndrome (ASS), which, in turn, causes male infertility (*Sudhakar et al., 2021*; *Tu et al., 2020*).

The sperm flagellum possesses an evolutionarily conserved axonemal structure composed of '9+2' microtubules, specifically, nine peripheral doublet microtubules (DMTs) surrounding two central microtubules known as the central pair (CP) (*Inaba and Mizuno, 2016*). Axonemal dyneins, radial spokes (RSs), and the nexin-dynein regulatory complex (N-DRC) are arranged on DMTs with a 96 nm repeating unit structures (*Kumar and Singh, 2021*). Within the axoneme, the N-DRC and RS are crucial for maintaining axonemal integrity, forming crossbridges between adjacent DMTs and linking the DMTs to the central apparatus, respectively (*Canty et al., 2021*; *Ishikawa, 2017*; *Kumar and Singh, 2021*). Recent advancements in artificial intelligence, biochemical techniques, and cryo-electron microscopy (cryo-EM) facilitated the analysis of the axonemal structures, revealing numerous components among RS, N-DRC, and DMTs that may serve as hubs for axoneme stabilization (*Bazan et al., 2021*; *Leung et al., 2023*; *Walton et al., 2023*; *Zhou et al., 2023*). For example, CFAP91 has been identified as a protein that extends from the base of RS2 through the N-DRC base plate to RS3, thus stabilizing RS2 and RS3 on the DMTs (*Bicka et al., 2022*; *Dymek et al., 2011*; *Gui et al., 2021*). Similarly, CFAP57 extends through the N-DRC and interacts with RS3 via its C-terminal region (*Ghanaeian et al., 2023*). Recent analyses suggest that CCDC96 and CCDC113 may form a complex that extends parallel to the N-DRC, connecting the base of RS3 to the tail of dynein g (IDA g) and the N-DRC (*Bazan et al., 2021*; *Ghanaeian et al., 2023*). However, the functions of these proteins in stabilizing the sperm flagellum remain unknown.

The sperm flagellum is tightly anchored to the sperm head through the HTCA, a complex structure based on the centrosome (*Wu et al., 2020*). This structure consists of two cylindrical microtubule-based centrioles and associated components, including well-organized segmented columns, capitulum plate, and basal plate. The segmented columns and capitulum plate, located below the basal plate, are thought to originate from the dense material emanating from the proximal centriole (*Fawcett and Phillips, 1969*; *Zamboni and Stefanini, 1971*). Many proteins have been identified in the sperm HTCA, with mouse models exhibiting phenotypes characteristic of ASS (*Wu et al., 2020*). SPATA6 is the first protein identified as a component of the HTCA using the knockout mouse model, and is crucial for its formation (*Yuan et al., 2015*). Deficiencies in SUN5 (*Elkhatib et al., 2017*; *Fang et al., 2018*; *Liu et al., 2020*; *Sha et al., 2018*; *Shang et al., 2018*; *Shang et al., 2017*; *Xiang et al., 2022*; *Zhang et al., 2021c*; *Zhu et al., 2016*) and PMFBP1 (*Deng et al., 2022*; *Liu et al., 2020*; *Liu et al., 2021*; *Lu et al., 2021*; *Nie et al., 2022*; *Sha et al., 2019*; *Zhu et al., 2018*) have been associated with ASS in both humans and mice. The centriole-related protein CENTLEIN acts as a bona fide linker between SUN5 and PMFBP1, participating in the HTCA assembly (*Zhang et al., 2021c*). Notably, impaired HTCA often coincides with defects in the sperm flagellum (*Hall et al., 2013*; *Shang et al., 2017*; *Yuan et al., 2015*; *Zhang et al., 2021c*; *Zhu et al., 2018*), suggesting that the stabilization of the HTCA may be closely associated with the integrity of sperm flagellum. However, the mechanism that maintains the stabilization of both the sperm flagellum and the HTCA remains to be clarified.

Here, we identified an evolutionarily conserved coiled-coil domain-containing (CCDC) protein, CCDC113, and found that it forms a complex with CFAP57 and CFAP91, thereby facilitating the connection of RS, N-DRC, and DMTs in the axoneme. Knockout of *Ccdc113* resulted in spermatozoa with flagellar defects and head-tail linkage detachment, leading to male infertility. Ultrastructural analysis showed that the loss of CCDC113 disrupted both the sperm axoneme and HTCA. CCDC113 localizes on the manchette, HTCA, and flagellum in elongating and elongated spermatids. Further analysis revealed that CCDC113 is indispensable for the connection of CFAP91 and DRC2 with DMTs in the sperm axoneme, and it interacts with SUN5 and CENTLEIN to stabilize the sperm HTCA. These results suggest that CCDC113 serves as a critical hub in maintaining the structural integrity of both the sperm flagellum and HTCA.

# Results

## CCDC113 complexes with CFAP57 and CFAP91

CCDC113 is an evolutionarily conserved CCDC protein identified in the ciliated species. Comparative analysis of CCDC113 structures from *Tetrahymena thermophila* to *Homo sapiens* showed structural similarity among CCDC113 orthologs (*Figure 1A*). Recent cryo-EM analysis in the structure of the 96 nm modular repeats of axonemes from the *T. thermophila* cilia and human respiratory cilia revealed that CCDC113 localizes to the linker region among RS, N-DRC, and DMTs (*Figure 1B*), suggesting it may serve as a structural component connecting RS, N-DRC, and DMTs (*Bazan et al., 2021*; *Ghanaeian et al., 2023*). To further investigate, we examined the interactions between CCDC113 and its neighboring axoneme-associated proteins, CFAP57 and CFAP91 (*Figure 1B*). We transfected HEK293T cells with a GFP-tagged CCDC113 and FLAG-tagged CFAP57 or CFAP91, then performed anti-FLAG-immunoprecipitations. CCDC113 was present in both FLAG-CFAP57 and FLAG-CFAP91 immunoprecipitates (*Figure 1C and D*), indicating CCDC113 interacts with both CFAP57 and CFAP91. Given that CFAP91 has been reported to stabilize RS on the DMTs (*Bicka et al., 2022*; *Dymek et al., 2011*; *Gui et al., 2021*) and cryo-EM analysis shows that CCDC113 is closed to DMTs, we speculated that CCDC113 may connect RS to DMTs by binding to CFAP91 and microtubules. To test this, we detected the interaction between CCDC113 and β-tubulin (TUBB5) and found that CCDC113 was present in MYC-TUBB5 immunoprecipitate (*Figure 1E*). Additionally, since CFAP57 extends through the N-DRC and CCDC113 is closed to the N-DRC (*Ghanaeian et al., 2023*), we further examined the interaction between CCDC113 and N-DRC components adjacent to DMTs. Co-immunoprecipitation (co-IP) analysis showed that CCDC113 could bind to DRC1, DRC2, and DRC3 (*Figure 1F–H*). We also included two sperm outer dense fiber proteins, ODF1 and ODF2 (*Zhu et al., 2022*), which are located far from the sperm axoneme, as negative controls in the co-IP experiments. As shown in *Figure 1—figure supplement 1A and B*, neither ODF1 nor ODF2 bound to CCDC113. Therefore, CCDC113 may function as an adaptor protein connecting RS, N-DRC, and DMTs, serving as a critical hub for axoneme stabilization.

## CCDC113 is required for male fertility

To investigate the physiological functions of CCDC113, we generated a *Ccdc113* knockout mouse strain using the CRISPR/Cas9 system (*Figure 2A*). The *Ccdc113*⁻/⁻ mice were genotyped by genomic DNA sequencing and further confirmed by polymerase chain reaction. Genotypes were distinguished by a 461 bp band for *Ccdc113*⁺/⁺ mice, a 539 bp band for *Ccdc113*⁻/⁻ mice, and two bands of 539 bp and 461 bp for *Ccdc113*⁺/⁻ mice (*Figure 2B*). Further immunoblotting analysis confirmed the elimination of CCDC113 in total protein extracts from *Ccdc113*⁻/⁻ testes (*Figure 2C*; *Figure 2—figure supplement 1A*). Mice lacking *Ccdc113* showed no gross abnormalities in their appearance or behavior, and no obvious differences in body weight (*Figure 2H*). Additionally, no hydrocephalus or left-right asymmetry defects were observed (*Figure 2—figure supplement 1B*). Additionally, the deficiency of CCDC113 did not affect ciliogenesis in the lung and trachea (*Figure 2—figure supplement 1C–E*). We then assessed the fertility of 2-month-old male and female *Ccdc113*⁻/⁻ mice. *Ccdc113*⁻/⁻ female mice were able to produce offspring after mating with wild-type (WT) adult males, similar to *Ccdc113*⁺/⁺ female mice (*Figure 2D*). However, *Ccdc113*⁻/⁻ male mice exhibited normal mating behavior, as indicated by the presence of copulatory plugs, they failed to produce offspring when mating with WT adult female mice (*Figure 2E*). Thus, the knockout of *Ccdc113* results in male infertility.

## *Ccdc113* knockout mice produce spermatozoa with flagellar defects and head-tail linkage detachment

To further investigate the cause of male infertility, we initially examined *Ccdc113*⁻/⁻ testis at both macroscopic and histological levels. *Ccdc113* knockout did not affect testis size (*Figure 2F and G*) or the ratio of testis weight to body weight (*Figure 2H and I*). Histological sections stained with hematoxylin-eosin (H&E) revealed that seminiferous tubules of *Ccdc113*⁺/⁺ mice exhibited a tubular lumen with flagella emerging from the developing spermatids. In contrast, flagellar staining appeared reduced in *Ccdc113*⁻/⁻ seminiferous tubules (*Figure 2J*, red asterisks). Immunofluorescence staining for acetylated tubulin (ac-TUBULIN), a marker for sperm flagellum (*Martinez et al., 2020*), further confirmed the flagellar defects in *Ccdc113*⁻/⁻ mice (*Figure 2K and L*).

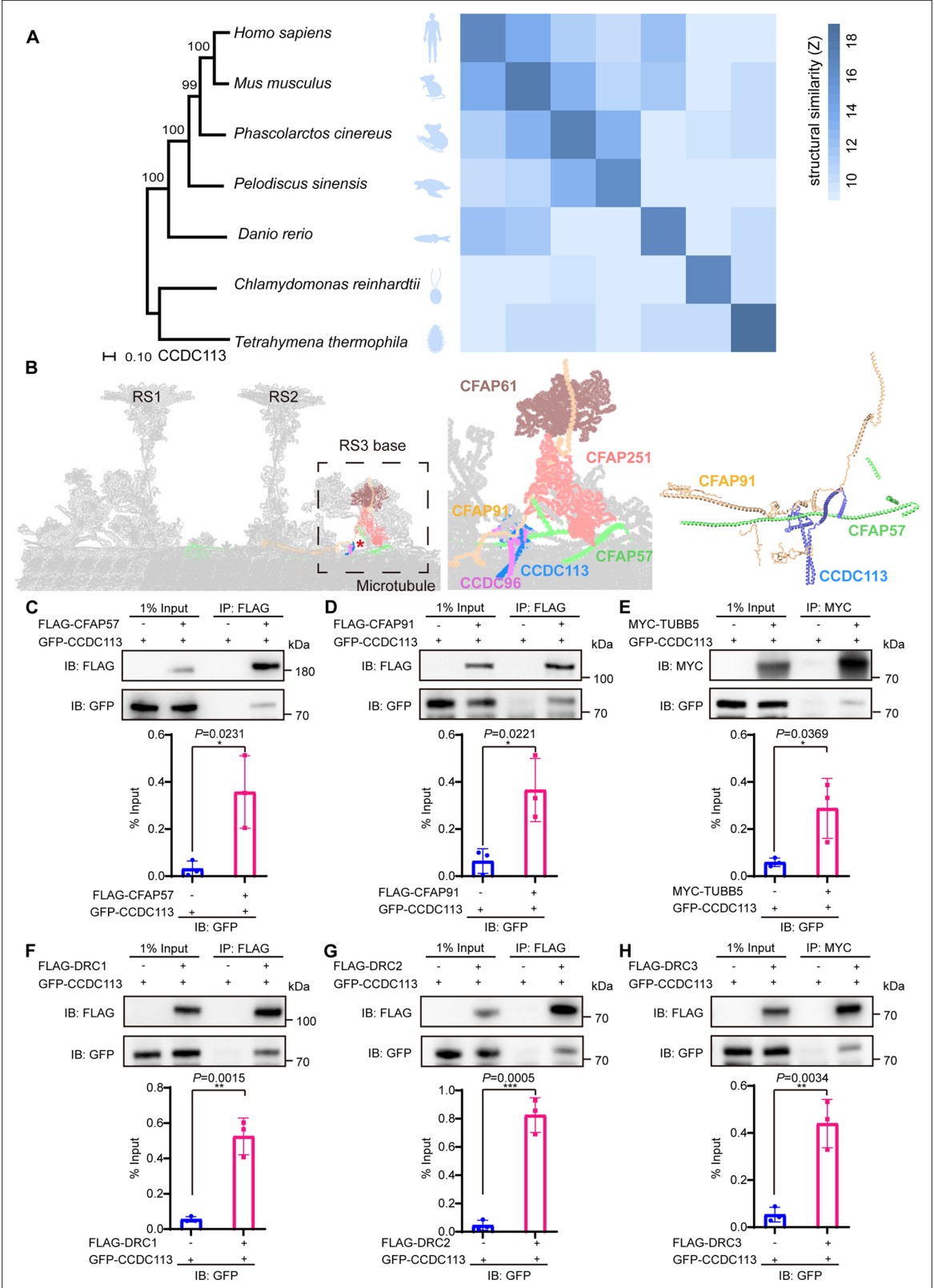

**Figure 1.** CCDC113 is an evolutionarily conserved axoneme-associated protein. (**A**) Multiple species phylogenetic tree of CCDC113. Structural similarity scores (Z scores) of CCDC113 orthologs in *H. sapiens*, *Mus musculus*, *Phascolarctos cinereus*, *Danio rerio*, *Chlamydomonas reinhardtii*, and *T. thermophila* were derived through the DALI webserver for pairwise structure comparisons (***Holm and Laakso, 2016***). (**B**) Positioning of CCDC113 within the 96 nm repeat of human axoneme (***Walton et al., 2023***). CCDC113 forms a complex with CCDC96, is located at the base of RS3, and is adjacent to

*Figure 1 continued on next page*

*Figure 1 continued*

CFAP91 and CFAP57. CFAP91 originates at the base of RS2 and links the RS3 subunits (CFAP251 and CFAP61). (**C–H**) Neighboring axoneme-associated proteins were expressed alone or co-expressed with CCDC113 in HEK293T cells, and the interactions between CCDC113 and CFAP57, CFAP91, TUBB5, DRC1, DRC2, or DRC3 were examined by co-immunoprecipitation. IB: immunoblotting; IP: immunoprecipitation. The % Input is displayed below the corresponding figures for quantification. n=3 independent experiments. Data are presented as mean ± SD; *p<0.05, **p<0.01, ***p<0.001.

The online version of this article includes the following source data and figure supplement(s) for figure 1:

**Source data 1.** Original files for western blot in *Figure 1C-H*.

**Source data 2.** Labelled files for western blot in *Figure 1C-H*.

**Figure supplement 1.** CCDC113 could not bind to ODF1 and ODF2.

**Figure supplement 1—source data 1.** Original files for western blot in *Figure 1—figure supplement 1A-B*.

**Figure supplement 1—source data 2.** Labelled files for western blot in *Figure 1—figure supplement 1A-B*.

Subsequently, we examined spermatids at different stages in *Ccdc113*$^{-/-}$ testes using periodic acid Schiff (PAS) staining. Pioneering work in the mid-1950s used the PAS staining in histological sections of mouse testis to visualize glycoproteins of the acrosome and Golgi in seminiferous tubules (*Oakberg, 1956*). The pioneers discovered in cross-sectioned seminiferous tubules the association of differentiating germ cells with successive layers to define different stages that in mice are 12, indicated as Roman numerals (XII). For each stage, different associations of maturing germ cells were always the same with early cells in differentiation at the periphery and more mature cells near the lumen. In this way, progressive differentiation from stem cells to mitotic, meiotic, acrosome-forming, and post-acrosome maturing spermatocytes was mapped to define spermatogenesis with the XII stages in mice representing the seminiferous cycle. The maturation process from acrosome-forming cells to mature spermatocytes is defined as spermiogenesis with 16 different steps that are morphologically distinct spermatids (*O'Donnell, 2014*). While acrosome biogenesis and nuclear morphology in *Ccdc113*$^{-/-}$ spermatids from steps 1–10 were comparable to those in *Ccdc113*$^{+/+}$ spermatids, abnormal club-shaped heads were observed in spermatids from steps 11–16 in *Ccdc113*$^{-/-}$ mice (*Figure 3—figure supplement 1A*, black asterisk). In addition, the manchette of *Ccdc113*$^{-/-}$ spermatids was more elongated compared to that of *Ccdc113*$^{+/+}$ spermatids (*Figure 3—figure supplement 1B*). Therefore, the disruption of CCDC113 impaired spermiogenesis.

Next, we examined the spermatozoa in the cauda epididymis and found that the sperm count in the *Ccdc113*$^{-/-}$ cauda epididymis was significantly decreased compared to the control group (*Figure 3A and B*). The motility of the released spermatozoa from *Ccdc113*$^{+/+}$ and *Ccdc113*$^{-/-}$ cauda epididymis showed that *Ccdc113*$^{-/-}$ spermatozoa were completely immotile (*Figure 3C*). H&E staining of the cauda epididymis showed fewer hematoxylin-stained sperm heads in the *Ccdc113*$^{-/-}$ cauda epididymis compared to the *Ccdc113*$^{+/+}$ cauda epididymis. Notably, unlike the control group, which exhibited linear eosin staining in the epididymal lumen, the *Ccdc113*$^{-/-}$ mice showed numerous coiled eosin-stained structures without sperm heads in the epididymal lumen (*Figure 3A*, red circles). To determine the morphological characteristics of the spermatozoa, we conducted single-sperm immunofluorescence using an anti-α/β-tubulin antibody to label the sperm flagellum and lectin peanut agglutinin (PNA) to visualize the sperm acrosome (*Nakata et al., 2015*). We noticed that *Ccdc113*$^{-/-}$ spermatozoa showed severe morphological malformations, including sperm head-tail detachment (type 1), abnormal sperm head with curly tail (type 2), normal sperm head with curly tail (type 3) (*Figure 3D and E*). To determine the role of CCDC113 in flagellum assembly, we analyzed flagellum formation during spermiogenesis. We found that in *Ccdc113*$^{-/-}$ mice, diffuse and curly axonemal signals were observed in testicular germ cells starting from the round spermatid stage (*Figure 3F*). Overall, these findings suggest that the deletion of CCDC113 leads to sperm flagellum deformities and detachment of the sperm head-to-tail linkage, resulting in a unique type of acephalic spermatozoa, which may be responsible for the *Ccdc113*$^{-/-}$ male infertility.

## CCDC113 localizes on the sperm neck and flagellum regions

To gain further insights into the functional role of CCDC113 during spermiogenesis, we examined its expression in different tissues and found that CCDC113 was predominantly expressed in mouse testis (*Figure 4—figure supplement 1A and B*). CCDC113 was first detected in testis at postnatal day 7 (P7), with expression levels increasing continuously from P21 onward, peaking in adult testes

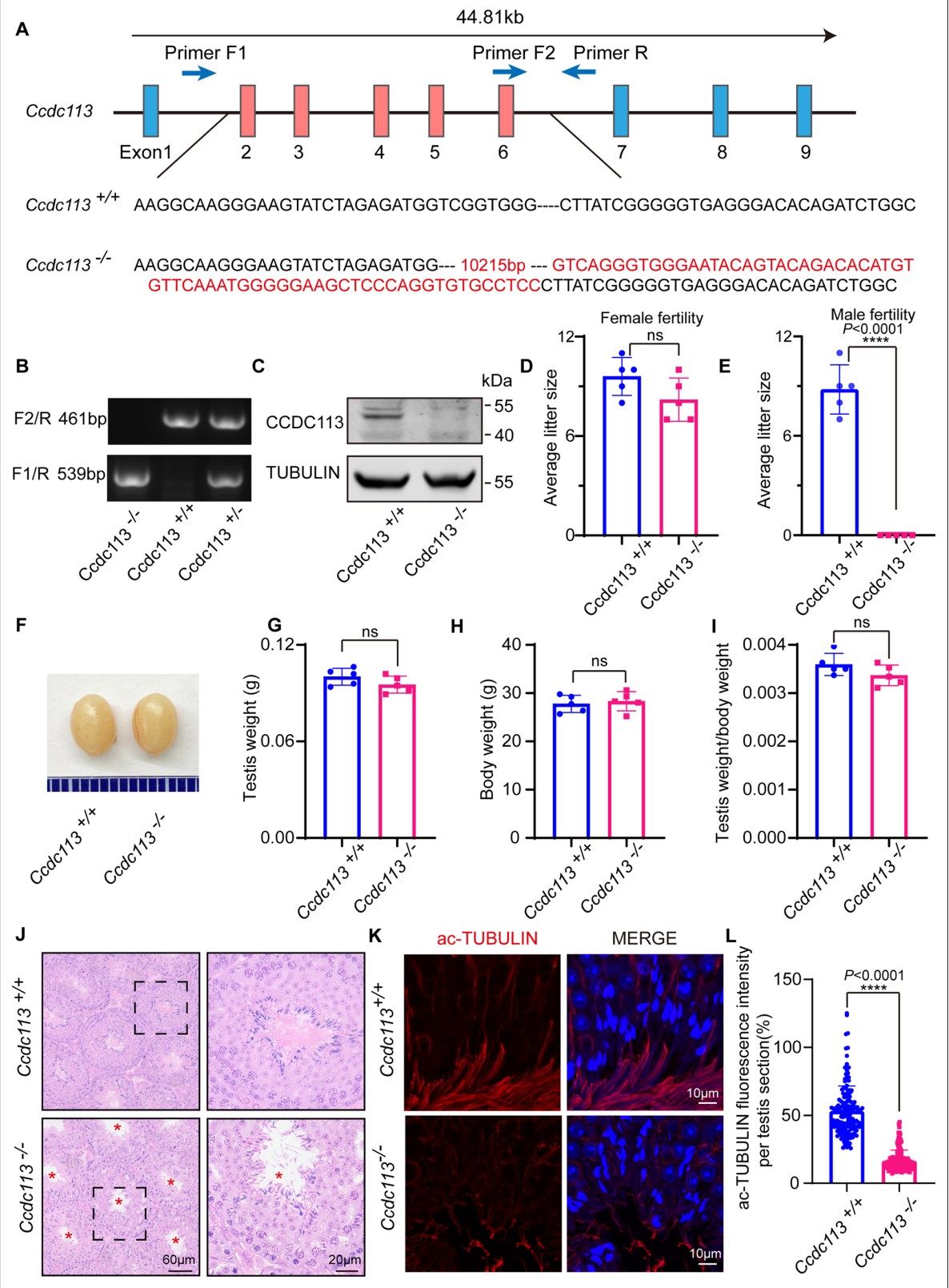

**Figure 2.** *Ccdc113* knockout leads to male infertility. (**A**) The CRISPR-Cas9 strategy for generating the *Ccdc113* knockout mice. (**B**) Genotyping to identify *Ccdc113* knockout mice. (**C**) Immunoblotting of CCDC113 in *Ccdc113⁺/⁺* and *Ccdc113⁻/⁻* testes. TUBULIN served as the loading control. (**D**) The average litter size of *Ccdc113⁺/⁺* and *Ccdc113⁻/⁻* female mice in 2 months (n=5 independent experiments). Data are presented as mean ± SD; ns indicates no significant difference. (**E**) The average litter size of *Ccdc113⁺/⁺* and *Ccdc113⁻/⁻* male mice in 2 months (n=5 independent experiments). Data

*Figure 2 continued*

are presented as mean ± SD; ****p<0.0001. (**F**) The size of testes was similar in *Ccdc113⁺/⁺* and *Ccdc113⁻/⁻* mice. (**G**) The testis weights of *Ccdc113⁺/⁺* and *Ccdc113⁻/⁻* male mice (n=5 independent experiments). Data are presented as mean ± SD; ns indicates no significant difference. (**H**) The body weights of *Ccdc113⁺/⁺* and *Ccdc113⁻/⁻* male mice (n=5 independent experiments). Data are presented as mean ± SD; ns indicates no significant difference. (**I**) The ratio of testis weight/body weight in *Ccdc113⁺/⁺* and *Ccdc113⁻/⁻* male mice (n=5 independent experiments). Data are presented as mean ± SD; ns indicates no significant difference. (**J**) Hematoxylin-eosin (H&E) staining of testis sections from *Ccdc113⁺/⁺* and *Ccdc113⁻/⁻* male mice. Red asterisks indicate the abnormal sperm flagellum in the *Ccdc113⁻/⁻* testis seminiferous tubule. (**K**) Immunofluorescence of acetylated-tubulin (red) in testis sections from *Ccdc113⁻/⁻* male mice showed flagellar defects. (**L**) Acetylated-tubulin fluorescence intensity was measured per testis section in 155 sections from 3 *Ccdc113⁺/⁺* mice and 153 sections from 3 *Ccdc113⁻/⁻* male mice. Data are presented as mean ± SD; ****p<0.0001.

The online version of this article includes the following source data and figure supplement(s) for figure 2:

**Source data 1.** Original file for gel in *Figure 2B*.

**Source data 2.** Labelled file for gel in *Figure 2B*.

**Source data 3.** Original files for western blot in *Figure 2C*.

**Source data 4.** Labelled file for western blot in *Figure 2C*.

**Figure supplement 1.** *Ccdc113⁻/⁻* mice did not exhibit the ciliopathies, such as hydrocephalus, situs inversus, and abnormal ciliogenesis of tracheal cilia.

**Figure supplement 1—source data 1.** Original files for western blot in *Figure 2—figure supplement 1A*.

**Figure supplement 1—source data 2.** Labelled file for western blot in *Figure 2—figure supplement 1A*.

(*Figure 4A*), suggesting that CCDC113 is highly expressed throughout spermiogenesis. We then conducted immunofluorescence analysis of CCDC113 in *Ccdc113⁺/⁺* and *Ccdc113⁻/⁻* germ cells to determine its precise localization during spermatogenesis. CCDC113 appeared as punctate signals near the nuclei of spermatocyte and round spermatids, and localized to the manchette, sperm neck, and flagellum regions in elongating and elongated spermatids (*Figure 4—figure supplement 1C*).

To further validate these findings, we co-stained CCDC113 with α/β-tubulin, which marks the manchette and flagellum in spermatids (*Lehti and Sironen, 2016*; *Figure 4B*). The immunofluorescence analysis showed that CCDC113 localized to the manchette surrounding the spermatid head from step 9 to step 14, as well as to the sperm neck and flagellum in the testes (*Figure 4B*). Given that CCDC113 was initially identified as a component of centriolar satellites (*Firat-Karalar et al., 2014*), the punctate signals of CCDC113 in spermatocyte and spermatids may be localized around the centrosome. To test this, we performed the immunofluorescent staining of CCDC113 and centrosomal protein CENTRIN1/2 in spermatocytes and spermatids, and found that the signal of CCDC113 partially colocalized with CENTRIN1/2 (*Figure 4C*). Thus, CCDC113 is localized to the centrosome, manchette, sperm neck, and flagellum regions in the developing germ cells.

Next, we examined the localization of CCDC113 in mature spermatozoa, and found that CCDC113 was localized in the sperm neck and flagellum regions (*Figure 4D*). Similar localization of CCDC113 was also observed in human mature spermatozoa (*Figure 4E*). The consistent localization of CCDC113 at the sperm neck and flagellum in mature spermatozoa suggests its importance for maintaining the integrity of the sperm flagellum and head-to-tail connection.

## *Ccdc113* knockout results in the disorganization of the sperm flagellum structures

To delineate the sperm flagellar defects in *Ccdc113⁻/⁻* mice, we conducted transmission electron microscopy (TEM) examination of longitudinal sections of *Ccdc113⁻/⁻* spermatozoa. TEM analysis revealed a significant presence of unremoved cytoplasm, including disrupted mitochondria, damaged axonemes, and large vacuoles in *Ccdc113⁻/⁻* spermatozoa (*Figure 5A*, red asterisks). Cross sections of the principal piece of *Ccdc113⁻/⁻* spermatozoa further revealed partial loss or unidentifiable '9+2' axonemal structures, along with the disruption of the fibrous sheath and outer dense fibers (*Figure 5A*). We further examined the axonemal structure in *Ccdc113⁻/⁻* testicular spermatids using TEM and found disorganized axonemal microtubules were detected in *Ccdc113⁻/⁻* testicular spermatids (*Figure 5B*). In contrast to the regular positioning of the CP and nine peripheral DMTs in the *Ccdc113⁺/⁺* spermatid axoneme, the *Ccdc113⁻/⁻* spermatids exhibited a scattered arrangement of DMTs, and no distinct RSs were observed (*Figure 5B*, red arrowheads). These results indicate CCDC113 is essential for the integrity of sperm flagellum.

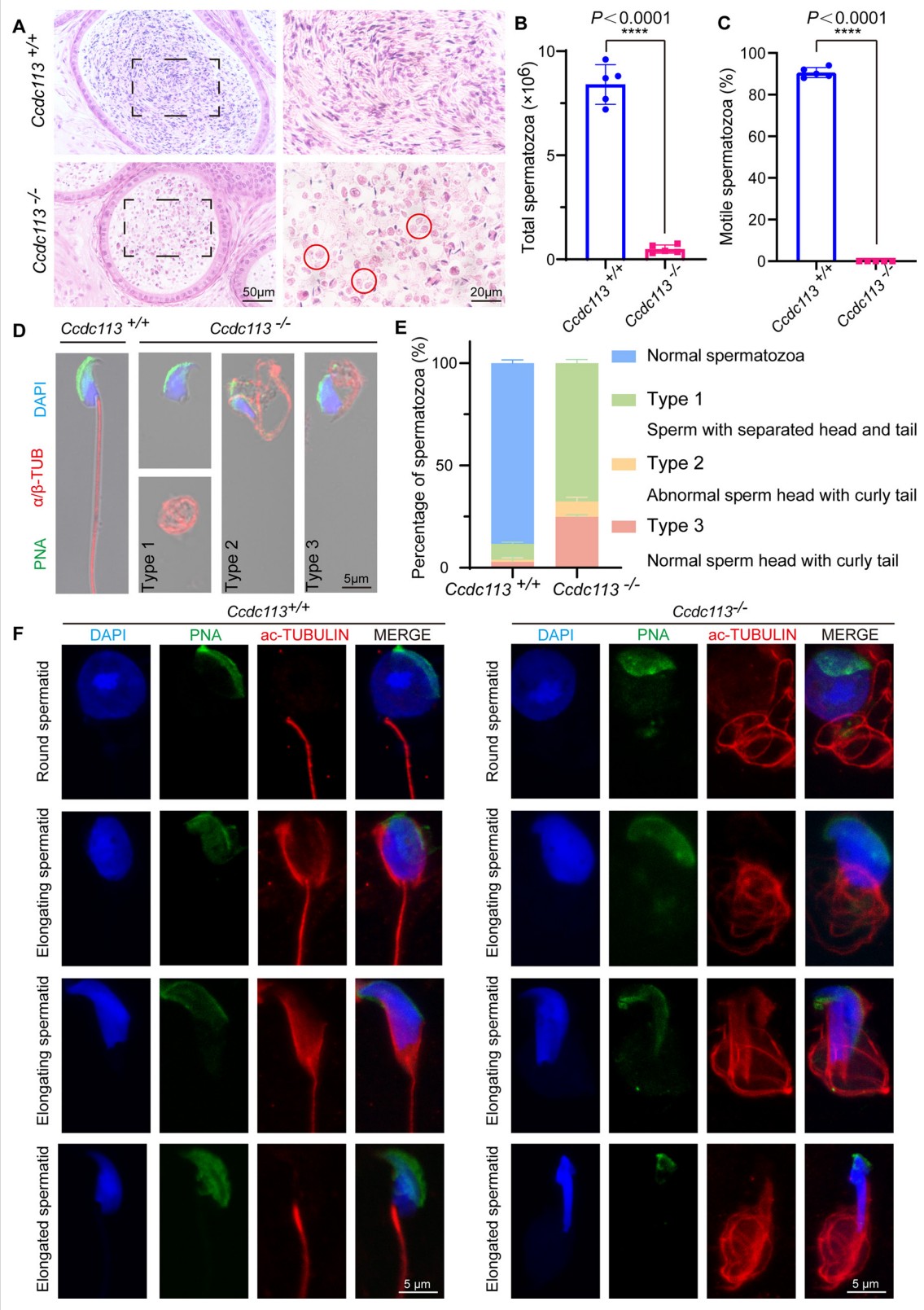

**Figure 3.** *Ccdc113* knockout results in sperm flagellar defects and sperm head-tail detachment. (**A**) Hematoxylin-eosin (H&E) staining of the cauda epididymis from 2-month-old *Ccdc113*+/+ and *Ccdc113*−/− male mice. Red circles indicate coiled eosin-stained structures without sperm heads in the epididymal lumen. (**B**) Analysis of sperm counts in *Ccdc113*+/+ and *Ccdc113*−/− male mice (n=5 independent experiments). Mature spermatozoa were extracted from the unilateral cauda epididymis and dispersed in phosphate-buffered saline (PBS). Sperm counts were measured using hemocytometers.

*Figure 3 continued on next page*

Figure 3 continued

Data are presented as mean ± SD; ****p<0.0001. (C) Motile sperm in *Ccdc113*$^{+/+}$ and *Ccdc113*$^{-/-}$ mice (n=5 independent experiments). Data are presented as mean ± SD; ****p<0.0001. (D) *Ccdc113*$^{+/+}$ and *Ccdc113*$^{-/-}$ spermatozoa were co-stained with a flagellar marker α/β-tubulin (red) and an acrosomal marker peanut agglutinin (PNA). Nuclei were stained with DAPI (blue). (E) Quantification of different categories of *Ccdc113*$^{+/+}$, *Ccdc113*$^{-/-}$ spermatozoa (n=3 independent experiments). Data are presented as mean ± SD. (F) Immunofluorescence analysis of acetylated-tubulin (green) and PNA (red) from *Ccdc113*$^{+/+}$ and *Ccdc113*$^{-/-}$ spermatids.

The online version of this article includes the following figure supplement(s) for figure 3:

**Figure supplement 1.** *Ccdc113* knockout leads to abnormal sperm head shaping defect.

CCDC113 has been shown to localize at the base of the RS3 and interact with adjacent axoneme-associated proteins (*Figure 1B–H*). Given the disorganized '9+2' axonemal structure was detected in cross-sectioned *Ccdc113*$^{-/-}$ flagellar specimen (*Figure 5A and B*), we speculated that CCDC113 likely served as an adaptor to connect the neighboring axoneme-associated proteins to DMTs. To test this, we examined the flagellar localization of CFAP91 in *Ccdc113*$^{-/-}$ spermatozoa, which is positioned in close proximity to CCDC113 at the root region of RS3 and is critical for the localizations of calmodulin-associated and spoke-associated complex proteins CFAP61 and CFAP251 (*Bicka et al., 2022*; *Meng et al., 2024*). Immunofluorescence results indicated that the absence of *Ccdc113* leads to the abnormal distribution of CFAP91 on the axoneme, where CFAP91 could not colocalize with DMTs (*Figure 5C*, white asterisks, E and F). Given that DRC2 serves as the core component of the axonemal N-DRC (*Jreijiri et al., 2024*) and CCDC113 could bind to DRC2 (*Figure 1G*), we further examined DRC2 localization in the *Ccdc113*$^{+/+}$ and *Ccdc113*$^{-/-}$ spermatozoa. Immunofluorescence analysis showed that DRC2 exhibited distinct signals that did not colocalize with DMTs of *Ccdc113*$^{-/-}$ spermatozoa (*Figure 5D*, white asterisks, G and H). These findings collectively indicate that CCDC113 is indispensable for the connection of CFAP91 and DRC2 to the DMTs, which is required for structural integrity of the sperm axoneme.

### *Ccdc113* knockout impairs head-to-tail anchorage of the spermatids

To explore the mechanism behind acephalic spermatozoa in *Ccdc113* knockout mice, we first examined where the flagellum detached from sperm head in *Ccdc113*$^{-/-}$ mice. The proportion of decapitated tails in the caput, corpus, and cauda of *Ccdc113*$^{-/-}$ epididymis was similar (*Figure 6A*), suggesting the separation of the sperm head and tail in *Ccdc113*$^{-/-}$ mice may occur either within the seminiferous tubules or upon entering the caput of the epididymis. To confirm this, we performed PAS staining to examine spermiogenesis stages in *Ccdc113*$^{+/+}$ and *Ccdc113*$^{-/-}$ testes (*Figure 6B*). We found that in *Ccdc113*$^{+/+}$ testes, sperm heads at stages VII–VIII were oriented toward the basal membrane, whereas in *Ccdc113*$^{-/-}$ testes, sperm heads were oriented toward the tubule lumen during these stages (*Figure 6C*, arrows and D), which may be due to the separation of sperm heads from the flagellum during spermiogenesis. Additionally, mature sperm heads were still present at stages IX–X in *Ccdc113*$^{-/-}$ testes, whereas mature spermatozoa were released into the lumen of the seminiferous tubule at stage VIII in *Ccdc113*$^{+/+}$ testes (*Figure 6B*, red asterisk). These results suggest that the sperm head and flagellum separation may occur during spermiation in the *Ccdc113*$^{-/-}$ testes.

Next, we examined the development of the HTCA in *Ccdc113*$^{+/+}$ and *Ccdc113*$^{-/-}$ spermatids using TEM. In *Ccdc113*$^{+/+}$ step 9–11 spermatids, the well-defined coupling apparatus, comprising the basal plate, capitulum plate, segmented columns, proximal centriole, distal centriole, was tightly attached to the sperm head. However, in *Ccdc113*$^{-/-}$ step 9–11 spermatids, the abnormal HTCA was detached from the sperm head (*Figure 6E*, red asterisk). Further observation of the HTCA structure revealed the absence of segmented columns and capitulum plate; only dense material surrounding the proximal centriole and basal plate could be detected (*Figure 6E*, white arrow). The basal plates were abnormally distant from their native implantation site on the nucleus of *Ccdc113*$^{-/-}$ elongating and elongated spermatids (*Figure 6E*). Taken together, our results indicate that the disruption of *Ccdc113* causes the destroyed coupling apparatus detachment from the sperm head during spermiogenesis, and CCDC113 is required for the integrity of the sperm HTCA.

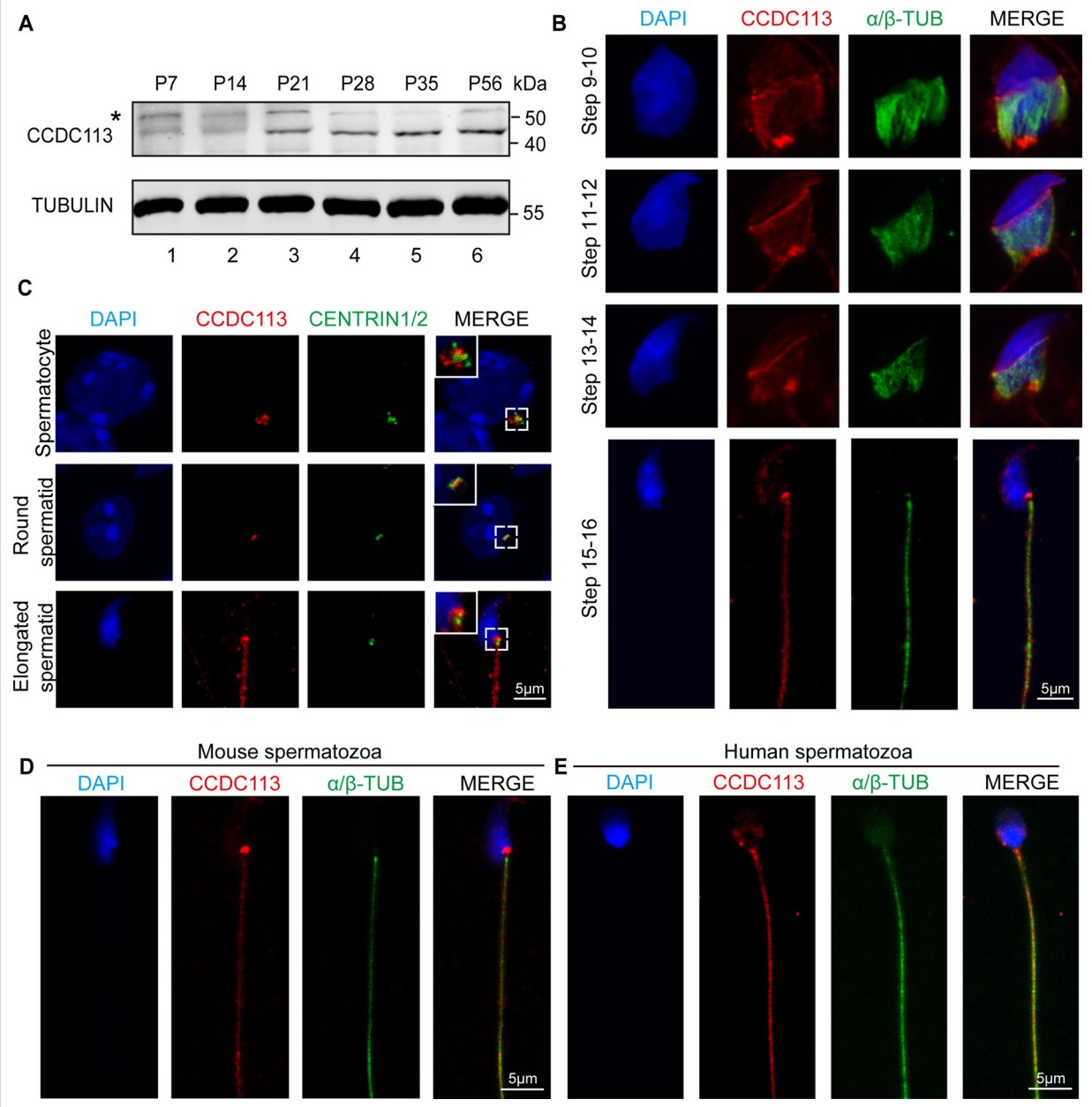

**Figure 4.** CCDC113 localizes to the head-tail coupling apparatus (HTCA), manchette, and sperm flagellum. (**A**) CCDC113 was expressed starting in postnatal day 7 (P7) testes. TUBULIN served as the loading control. An asterisk indicates nonspecific bands. (**B**) Immunofluorescence of CCDC113 (red) and CENTRIN1/2 (green) in developing germ cells. CCDC113 partially colocalize with centriolar protein CENTRIN1/2. (**C**) Immunofluorescence of CCDC113 (red) and α/β-tubulin (green) in developing germ cells. The manchette was stained with the anti-α/β-tubulin antibody. (**D–E**) CCDC113 localizes to the HTCA and flagellum in mature mouse spermatozoa and human spermatozoa. Nuclei were stained with DAPI (blue).

The online version of this article includes the following source data and figure supplement(s) for figure 4:

**Source data 1.** Original files for western blot in *Figure 4A*.

**Source data 2.** Labelled file for western blot in *Figure 4A*.

**Figure supplement 1.** Subcellular localization of CCDC113 in testicular germ cells from *Ccdc113⁺/⁺* and *Ccdc113⁻/⁻* mice.

**Figure supplement 1—source data 1.** Original files for western blot in *Figure 4—figure supplement 1B*.

**Figure supplement 1—source data 2.** Labelled file for western blot in *Figure 4—figure supplement 1B*.

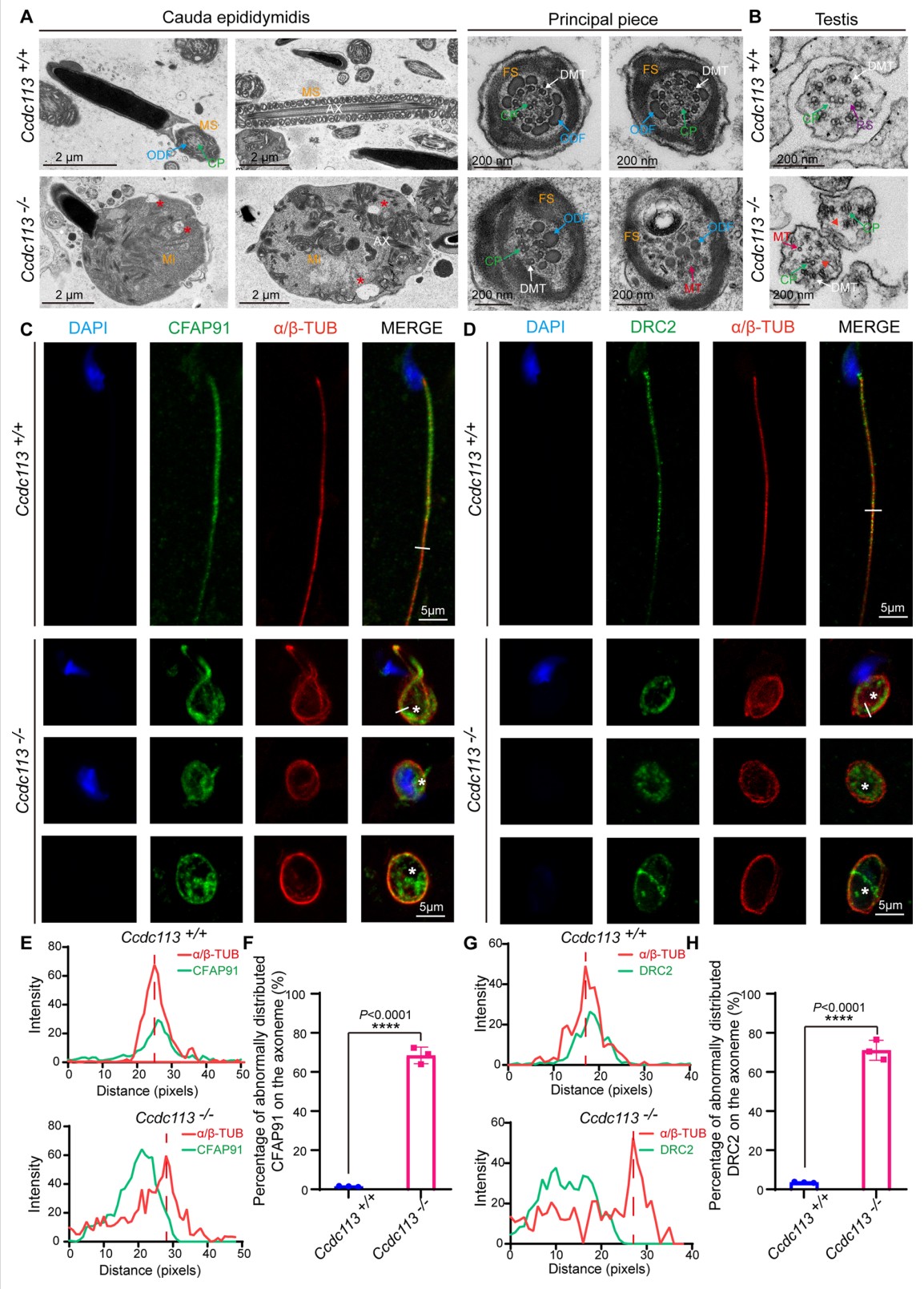

**Figure 5.** CCDC113 is indispensable for the docking of CFAP91 and DRC2 to the doublet microtubules (DMTs) to maintain the structural integrity of the axoneme. (**A**) Transmission electron microscopy (TEM) analysis of spermatozoa from the cauda epididymidis of *Ccdc113*^(+/+) and *Ccdc113*^(–/–) male mice. The flagellar longitudinal sections of *Ccdc113*^(–/–) spermatozoa revealed unremoved cytoplasm, including disrupted mitochondria, damaged axonemes, and large vacuoles. Asterisks indicate large vacuoles. Cross sections of the principal piece of *Ccdc113*^(–/–) spermatozoa further revealed partial loss or

*Figure 5 continued on next page*

*Figure 5 continued*

unidentifiable '9+2' structures, along with the disruption of the fibrous sheath and outer dense fibers. (**B**) TEM analysis of the axoneme in testicular spermatids from *Ccdc113*⁺/⁺ and *Ccdc113*⁻/⁻ male mice. The red arrowheads indicate the absence of significant radial spokes (RSs). MS: mitochondrial sheath; Mi: mitochondrial; AX: axoneme; FS: fibrous sheath; DMT: doublet microtubule; MT: microtubule; CP: central pair; ODF: outer dense fiber; RS: radial spokes. (**C**) The immunofluorescence analysis for CFAP91 (green) and α/β-tubulin (red) was performed in *Ccdc113*⁺/⁺ and *Ccdc113*⁻/⁻ spermatozoa. Nuclei were stained with DAPI (blue). White asterisks indicate regions not colocalized with tubulin. (**D**) The immunofluorescence analysis for DRC2 (green) and α/β-tubulin (red) was performed in *Ccdc113*⁺/⁺ and *Ccdc113*⁻/⁻ spermatozoa. Nuclei were stained with DAPI (blue). White asterisks indicate regions not colocalized with tubulin. (**E, G**) Line-scan analysis (white line) was performed using ImageJ software. (**F, H**) Percentage of abnormally distributed CFAP91 and DRC2 on the axoneme of *Ccdc113*⁺/⁺ and *Ccdc113*⁻/⁻ spermatozoa (n=3 independent experiments). At least 200 spermatozoa were analyzed from each mouse. Data are presented as mean ± SD; ****p<0.0001.

## CCDC113 cooperates with SUN5 and CENTLEIN to stabilize sperm HTCA

To elucidate the molecular function of CCDC113 in sperm head-tail linkage, we examined its interaction with known HTCA-associated proteins, including SUN5, CENTLEIN, PMFBP1, and SPATA6 (*Shang et al., 2017*; *Yuan et al., 2015*; *Zhang et al., 2021c*; *Zhu et al., 2018*). GFP-tagged CCDC113 and FLAG-tagged HTCA-associated proteins were co-transfected into HEK293T cells, followed by the immunoprecipitation with anti-GFP antibody. We found that CCDC113 interacted with SUN5 and CENTLEIN, but not with PMFBP1 and SPATA6 (*Figure 7A–C and F*). We further conducted co-IP assays in the reverse direction and verified that both SUN5 and CENTLEIN could bind to CCDC113 (*Figure 7D and E*). To investigate the localization of CCDC113 within the HTCA, we co-stained mature spermatozoa with antibodies against CCDC113 and SUN5, which has been reported to be localized at the root connecting the HTCA to the nuclear envelope (*Shang et al., 2017*; *Zhang et al., 2021a*). CCDC113 was found positioned below SUN5, showed partial overlap with it (*Figure 7G*). The centriolar protein CENTLEIN was localized to the HTCA and served as the critical linker protein between SUN5 and PMFBP1 in the elongating and elongated spermatids (*Zhang et al., 2021b*). Given that CENTLEIN disappears in mature spermatozoa (*Zhang et al., 2021c*), we performed the immunofluorescent staining of CCDC113 and CENTLEIN in elongated spermatids and found that CCDC113 partially colocalized with CENTLEIN at the HTCA (*Figure 7H*). Thus, CCDC113 interacts with both SUN5 and CENTLEIN and localizes at the sperm HTCA.

The HTCA localization of CCDC113 may be responsible for maintaining HTCA integrity. To test this hypothesis, we examined the localization of classical HTCA component SPATA6 in *Ccdc113*⁻/⁻ and *Ccdc113*⁺/⁺ spermatozoa using immunofluorescent staining. We found that SPATA6 was not attached to the implantation fossa of the sperm nucleus in *Ccdc113*⁻/⁻ spermatozoa (*Figure 7I and J*), indicating that CCDC113 is essential for the integrity of the sperm HTCA. To further elucidate the functional relationships among CCDC113, SUN5, CENTLEIN, and PMFBP1 at the sperm HTCA, we examined the localization of CCDC113 in *Sun5*⁻/⁻, *Centlein*⁻/⁻, and *Pmfbp1*⁻/⁻ spermatozoa. Compared to the control group, CCDC113 was predominantly localized on the decapitated flagellum in *Sun5*⁻/⁻, *Centlein*⁻/⁻, and *Pmfbp1*⁻/⁻ spermatozoa (*Figure 7K and L*), indicating SUN5, CENTLEIN, and PMFBP1 are crucial for the proper docking of CCDC113 to the implantation site on the sperm head. Taken together, these data demonstrate that CCDC113 cooperates with SUN5 and CENTLEIN to stabilize the sperm HTCA and anchor the sperm head to the tail.

## Discussion

CCDC113 is a highly evolutionarily conserved component of motile cilia/flagella. Studies in the model organism, *T. thermophila*, have revealed that CCDC113 connects RS3 to dynein g and the N-DRC, which plays essential role in cilia motility (*Bazan et al., 2021*; *Ghanaeian et al., 2023*). Recent studies have also identified the localization of CCDC113 within the 96 nm repeat structure of the human respiratory epithelial axoneme, and localizes to the linker region among RS, N-DRC, and DMTs (*Walton et al., 2023*). In this study, we reveal that CCDC113 is indispensable for male fertility, as *Ccdc113* knockout mice produce spermatozoa with flagellar defects and head-tail linkage detachment (*Figure 3D*). CCDC113 is localized to the sperm neck and flagellum regions in the elongating and elongated spermatids. In the sperm flagellum, CCDC113 interacts with both CFAP57 and CFAP91, serving as an adaptor protein to connect RS, N-DRC, and DMTs, thereby stabilizing sperm flagellum.

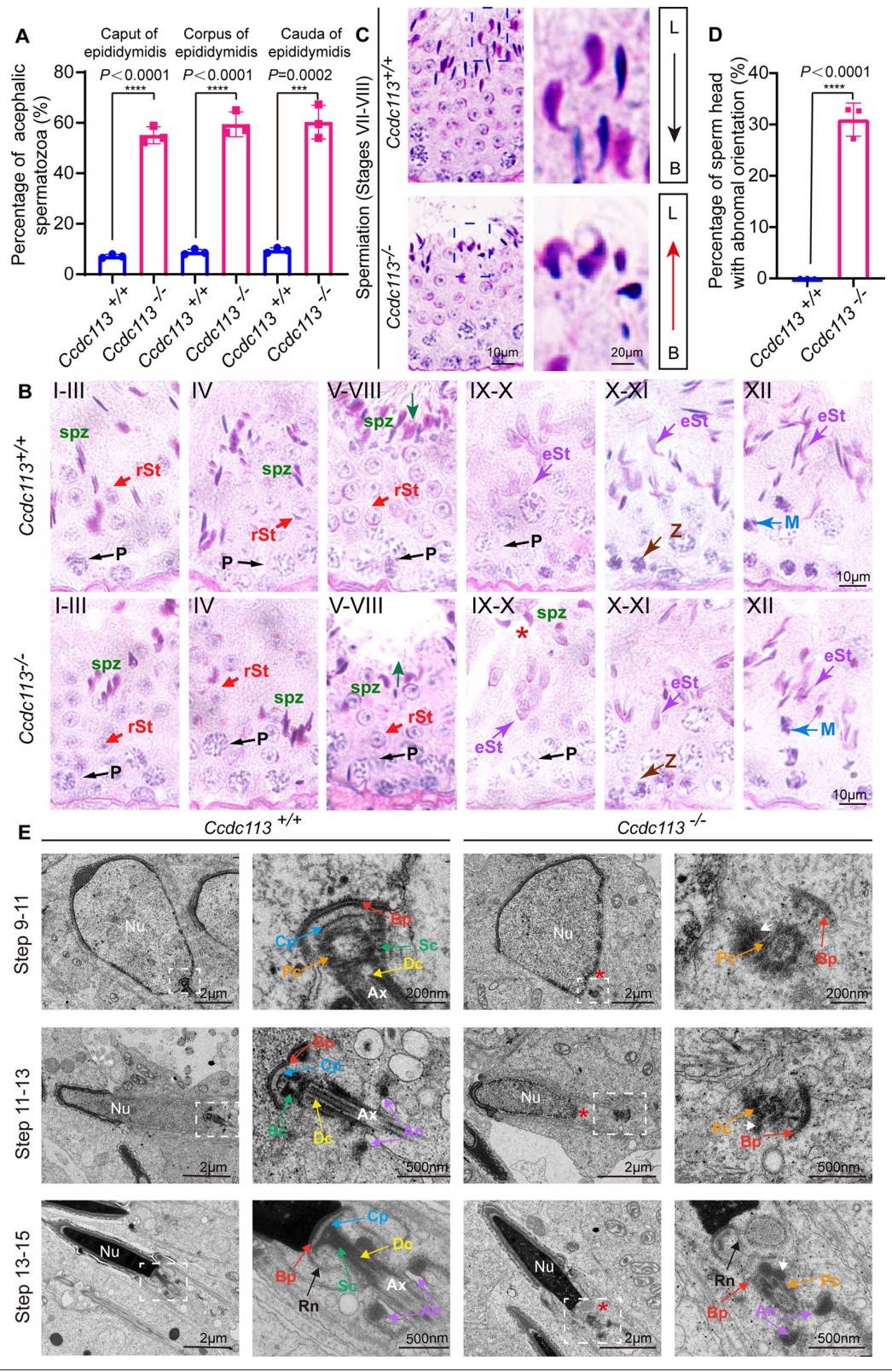

**Figure 6.** *Ccdc113* knockout spermatids display impaired head-tail coupling apparatus (HTCA). (**A**) The proportion of decapitated tails in *Ccdc113*⁺/⁺ and *Ccdc113*⁻/⁻ corpus, caput, and cauda epididymis (n=3 independent experiments). At least 200 spermatozoa were analyzed from each mouse. Data are presented as mean ± SD; ****p<0.0001, ***p<0.001. (**B**) Periodic acid Schiff (PAS) staining of testis sections from *Ccdc113*⁺/⁺ and *Ccdc113*⁻/⁻

*Figure 6 continued on next page*

*Figure 6 continued*

mice. The green arrows indicate the orientation of the sperm heads in stages V–VIII seminiferous epithelia. *Ccdc113⁻/⁻* sperm head could still be detected in stages IX–X seminiferous epithelia. P: pachytene spermatocyte, spz: spermatozoa, rSt: round spermatid, eSt: elongating spermatid, Z: zygotene spermatocyte, M: meiotic spermatocyte. (**C**) *Ccdc113⁻/⁻* spermatids lost their head orientation toward the basement membrane during spermiation in stages VII–VIII of the seminiferous epithelium. L: lumen, B: basement membrane. (**D**) Percentage of sperm heads with abnormal orientation in stages VII–VIII of the seminiferous epithelium in *Ccdc113⁺/⁺* and *Ccdc113⁻/⁻* mice (n=3 independent experiments). At least 200 spermatozoa were analyzed from each mouse. Data are presented as mean ± SD; ****p<0.0001. (**E**) Defective HTCA formation in *Ccdc113⁻/⁻* spermatids. Transmission electron microscopy (TEM) analysis of the stepwise development of the HTCA was performed in *Ccdc113⁺/⁺* and *Ccdc113⁻/⁻* testes. In *Ccdc113⁺/⁺* spermatids, the well-defined coupling apparatus was tightly attached to the sperm head. In *Ccdc113⁻/⁻* spermatids, segmented columns (Scs), the capitulum (Cp) were absent. The red asterisks indicate the distance between the sperm head and HTCA. The white arrows indicate the dense material surrounding the proximal centriole. Nu: nuclear; Bp: basal plate; Cp: capitulum; Sc: segmented column; Pc: proximal centriole; Dc: distal centriole; An: annulus; Ax: axoneme; Rn: redundant nuclear envelope.

At the sperm head to tail connecting piece, CCDC113 binds to SUN5 and CENTLEIN to stabilize sperm HTCA and anchor the sperm head to the tail. Thus, CCDC113 is essential for maintaining the integrity of both the sperm axoneme and the sperm HTCA. Moreover, TEM analysis detected excess residual cytoplasm in spermatozoa, including disrupted mitochondria, damaged axonemes, and large vacuoles, indicating defects in cytoplasmic removal in *Ccdc113⁻/⁻* mice (*Figure 5A*).

Recent cryo-EM analysis of the axonemes from *T. thermophila* cilia and human respiratory cilia revealed that CCDC113 is localized at the linker region between RS3, N-DRC, and DMTs (*Bazan et al., 2021*; *Ghanaeian et al., 2023*). We also found that CCDC113 interacts with adjacent axoneme-associated proteins CFAP57 and CFAP91 (*Figure 1C and D*). CFAP57 has been identified as the adaptor protein responsible for assembling dynein g and d, and it can interact with both N-DRC and RS3 (*Ghanaeian et al., 2023*; *Ma et al., 2023*). Previous studies have demonstrated that CFAP91 extends from the base of RS2 through the N-DRC base plate to RS3, playing a crucial role in stabilizing and localizing RS2 and RS3 on the DMT (*Bicka et al., 2022*; *Dymek et al., 2011*; *Gui et al., 2021*). The CFAP91 ortholog, FAP91, interacts with three N-DRC subunits (DRC1, DRC2, and DRC4), facilitating the docking of the N-DRC in *Chlamydomonas* (*Gui et al., 2021*). In humans, pathogenic mutations in CFAP91 and DRC2 disrupt sperm flagellum structure and result in MMAF (*Jreijiri et al., 2024*; *Martinez et al., 2020*). TEM and immunostaining experiments in spermatozoa showed severe CP and RSs defects in *CFAP91* mutant patients (*Martinez et al., 2020*). In this study, we found that the absence of CCDC113 results in severe axonemal disorganization characterized by defective RSs, scattered DMTs, and misplaced CP. Further analysis demonstrated that the CCDC113 deficiency disrupts the localization of CFAP91 and DRC2 on DMTs. Thus, CCDC113 may function as an adaptor protein to stabilize CFAP91 and DRC2 on DMTs, facilitating the docking of RS and N-DRC to DMTs and thereby maintaining the integrity of sperm axoneme.

Recent analyses have revealed that certain centrosomal proteins play crucial roles in the assembly and maintenance of sperm HTCA (*Avasthi et al., 2013*; *Liska et al., 2009*; *Zhang et al., 2021c*). In early round spermatids, the centriole pair initially localizes to the caudal nuclear pole and expands the electron-dense material, some of which exhibits striation (*Wu et al., 2020*). As spermatids develop, the dense material around the centrioles gradually transforms into a well-organized structure, clearly identified as the basal plate, capitulum plate, and segmented columns (*Dooher and Bennett, 1973*; *Wu et al., 2020*). CCDC113 was initially identified through the isolation of centriolar proteins from bovine sperm (*Firat-Karalar et al., 2014*). During spermiogenesis, CCDC113 colocalizes with CENTRIN1/2 at the centrosome in developing spermatids and continues to localize to the sperm neck region in elongating and elongated spermatids (*Figure 4B and C*). Additionally, CCDC113 can bind to HTCA-associated centrosomal protein, CENTLEIN, and the disruption of *Centlein* impairs the attachment of CCDC113 to the sperm head. In *Ccdc113⁻/⁻* spermatids, the capitulum plate and segmented columns are absent, and the basal plate is detached from the implantation site on the nucleus of *Ccdc113⁻/⁻* elongating and elongated spermatids during spermiogenesis (*Figure 6E*). These observations raise the possibility that CCDC113 is an HTCA-associated centrosomal protein crucial for maintaining the structural integrity of the HTCA.

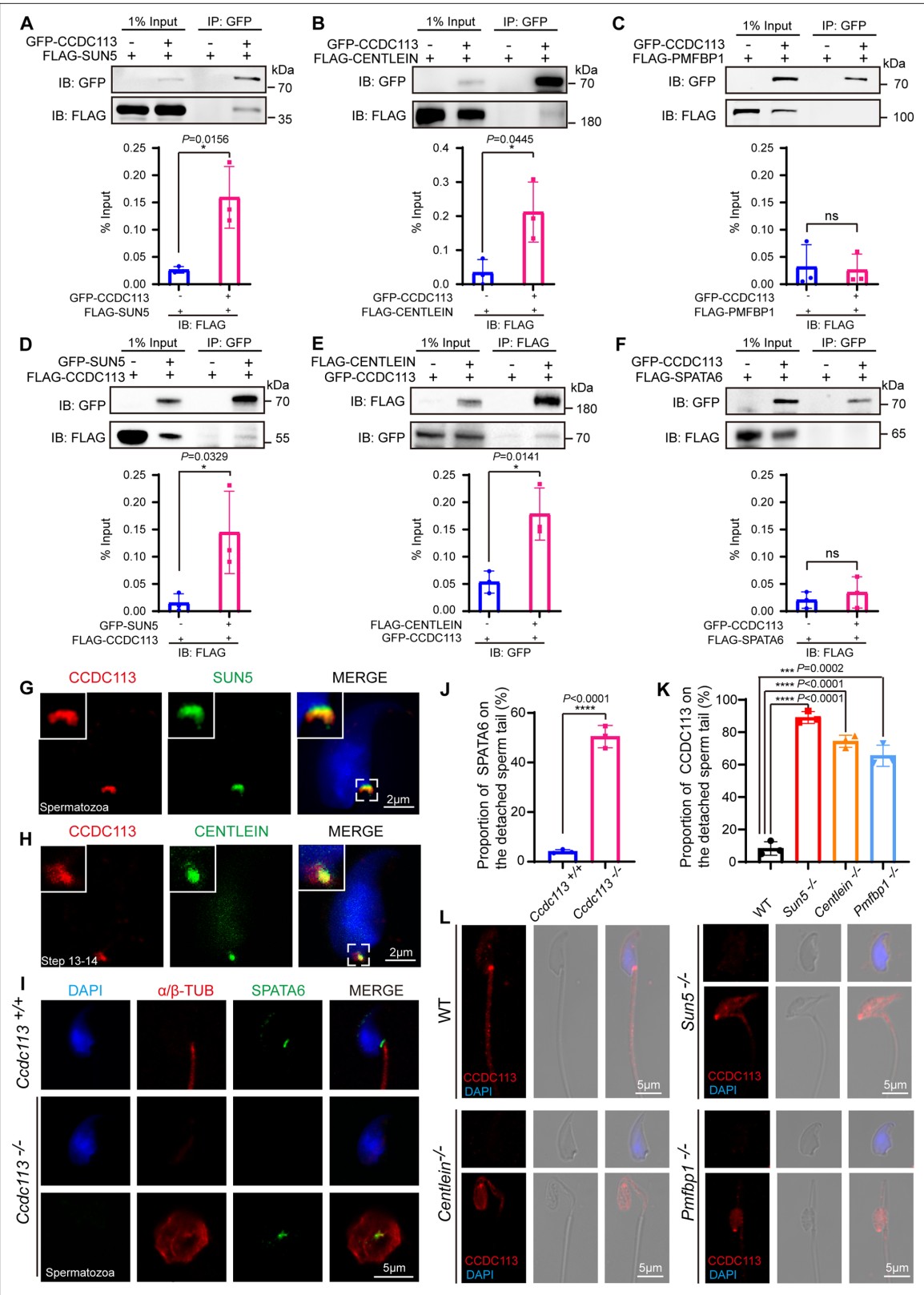

**Figure 7.** CCDC113 interacts with SUN5 and CENTLEIN, participating in sperm head-tail linkage. (**A–C, F**) Head-tail coupling apparatus (HTCA)-associated proteins (SUN5, CENTLEIN, PMFBP1, SPATA6) were expressed alone or co-expressed with CCDC113 in HEK293T cells, and the interactions between CCDC113 and these HTCA-associated proteins were examined by co-immunoprecipitation. CCDC113 interacted with SUN5 and CENTLEIN, but did not interact with PMFBP1 and SPATA6. IB: immunoblotting; IP: immunoprecipitation. (**D**) SUN5 interacted with CCDC113. pECMV-FLAG-

*Figure 7 continued on next page*

*Figure 7 continued*

*Ccdc113* and pEGFP-GFP-*Sun5* were transfected into HEK293T cells. At 48 hr after transfection, the cells were collected for immunoprecipitation (IP) with anti-GFP antibody and analyzed with anti-FLAG and anti-GFP antibodies. (**E**) CENTLEIN interacted with CCDC113. pCDNA -FLAG-*Centlein* and pEGFP-GFP-*Ccdc113* were transfected into HEK293T cells. At 48 hr after transfection, the cells were collected for IP with anti-FLAG antibody and analyzed with anti-FLAG and anti-GFP antibodies. The % Input is displayed below the corresponding figures for quantification. n=3 independent experiments. Data are presented as mean ± SD; *p<0.05, ns indicates no significant difference. (**G**) Immunofluorescence of CCDC113 (red) and SUN5 (green) in mature spermatozoa. Nuclei were stained with DAPI (blue). (**H**) Immunofluorescence of CCDC113 (red) and CENTLEIN (green) in testicular step 13–14 spermatid. Nuclei were stained with DAPI (blue). (**I**) Immunofluorescence analysis for SPATA6 (green) and α/β-tubulin (red) was performed in *Ccdc113*$^{+/+}$ and *Ccdc113*$^{-/-}$ spermatozoa. Nuclei were stained with DAPI (blue). (**J**) Quantification ratio of SPATA6 on the detached sperm tail (n=3 independent experiments). At least 200 spermatozoa were analyzed for each mouse. (**K**) Quantification ratio of CCDC113 on the detached sperm tail (n=3 independent experiments). At least 200 spermatozoa were analyzed from each mouse. Data are presented as mean ± SD; ***p<0.001, ****p<0.0001. (**L**) Immunofluorescence analysis for CCDC113 (red) was performed in wild-type (WT), *Sun5*$^{-/-}$, *Centlein*$^{-/-}$, and *Pmfbp1*$^{-/-}$ spermatozoa. Nuclei were stained with DAPI (blue).

The online version of this article includes the following source data for figure 7:

**Source data 1.** Original files for western blot in *Figure 7A-F*.

**Source data 2.** Labelled files for western blot in *Figure 7A-F*.

SUN5 is a transmembrane protein located in the nuclear envelope, and acts as the root connecting the HTCA to the sperm nuclear envelope (*Shang et al., 2017*). CENTLEIN can directly bind to both SUN5 and PMFBP1, serving as a linker between SUN5 and PMFBP1 to maintain the integrity of HTCA (*Zhang et al., 2021c*). CCDC113 can interact with SUN5 and CENTLEIN, but not PMFBP1 (*Figure 7A–C*), and left on the tip of the decapitated tail in *Sun5*$^{-/-}$ and *Centlein*$^{-/-}$ spermatozoa (*Figure 7K and L*). Furthermore, CCDC113 colocalizes with SUN5 in the HTCA region, and immunofluorescence staining in spermatozoa shows that SUN5 is positioned closer to the sperm nucleus than CCDC113 (*Figure 7G*). Therefore, SUN5 and CENTLEIN may be closer to the sperm nucleus than CCDC113. PAS staining revealed that *Ccdc113*$^{-/-}$ sperm heads are abnormally oriented in stages V–VIII seminiferous epithelia (*Figure 6C and D*), and TEM analysis further demonstrated that the disruption of CCDC113 causes the detachment of the destroyed coupling apparatus from the sperm head in step 9–11 spermatids (*Figure 6E*). All these results suggest that the detachment of sperm head and tail in *Ccdc113*$^{-/-}$ mice may not be a secondary effect of sperm flagellum defects.

Overall, we identified CCDC113 as a structural component of both the flagellar axoneme and the HTCA, where it performs dual roles in stabilizing the sperm axonemal structure and maintaining the structural integrity of HTCA. Given that the cryo-EM of sperm axoneme and HTCA could powerfully strengthen the role of CCDC113 in stabilizing sperm axoneme and HTCA, it represents valuable direction for future research. *Ccdc113*$^{-/-}$ mice did not exhibit other ciliopathies, such as situs inversus, hydrocephalus, or abnormal ciliogenesis of tracheal cilia (*Figure 2—figure supplement 1B–E*), which suggests that CCDC113 may specifically function in spermiogenesis.

# Materials and methods

## Phylogenetic analysis and structural similarity analysis

The amino acid sequence of CCDC113 of seven species were downloaded from UniProt. The phylogenetic trees were constructed using MEGA 10.0 (*Kumar et al., 2018*) with the neighbor-joining method (*Saitou and Nei, 1987*). 3D structures of CCDC113 orthologs of seven species were obtained from AlphaFold Protein Structure Database (*Varadi et al., 2022*). Structural similarity Z scores were derived through the DALI webserver for all against all structure comparison (*Holm and Laakso, 2016*).

## Animals

The *Sun5*$^{-/-}$, *Pmfbp1*$^{-/-}$, and *Centlein*$^{-/-}$ mice have been reported previously (*Shang et al., 2017*; *Zhang et al., 2021c*; *Zhu et al., 2018*). The mouse *Ccdc113* gene is 44.81 kb and contains 9 exons. Exon 2 to exon 6 of *Ccdc113* were chosen as the target sites. The knockout mice were generated using CRISPR-Cas9 system from Cyagen Biosciences. The gRNA and Cas9 mRNA were co-injected into fertilized eggs of C57BL/6J mice to generate *Ccdc113*$^{+/-}$ mice with a 10,215 bp base deletion and 64 bp insertion. The resulting heterozygotes were interbred, and their offspring were genotyped by genomic DNA sequencing to identify wide-type and homozygote mice. The genotyping primers

for knockout were: F1: 5'-TCAAATCATCACACCCTGCCTCT-3', R: 5'-GCTTGCACTCGGGTGATACA TAA-3', and for WT mice, the specific primers were: F2: 5'-CAGGTTCTCAACACCTACAAAGTA-3', R: 5'-GCTTGCACTCGGGTGATACATAA-3'.

All the animal experiments were performed according to approved Institutional Animal Care and Use Committee (IACUC) protocols (# 08-133) of the Institute of Zoology, Chinese Academy of Sciences.

## Cell line

HEK293T cells (Cat# GNHu43) were obtained from the Stem Cell Bank of the Chinese Academy of Sciences, authenticated by short tandem repeat profiling, and tested negative for mycoplasma contamination.

## Assessment of fertility

Fertility was assessed in 2-month-old male mice of various genotypes. Each male was paired with two WT females (6–8 weeks of age), and vaginal plugs were checked each morning. Females with a vaginal plug were separated and housed individually, and their pregnancy outcomes were recorded. If a female did not give birth by day 22 postcoitus, it was considered not pregnant and euthanized for confirmation. Each male participated in six cycles of the breeding assay with different females.

## Human adult sperm sample preparation

The sperm donation candidates in this study were healthy young Chinese men. Each participant underwent a thorough medical examination and completed an extensive medical and social questionnaire to ensure the exclusion of individuals with genetic or significant medical issues, as outlined in the Basic Standard and Technical Norms of Human Sperm Bank published by the Chinese Ministry of Health. Individuals who smoked, abused drugs, or were heavy drinkers were also excluded from the study. Those who remained eligible signed informed consent forms for voluntary sperm donation and agreed to reside in Beijing for a minimum of 6 months. The sperm bank documented each participant's age, date of birth, and date of semen collection. Ethical approval for this study was granted by the Reproductive Study Ethics Committee of Peking University Third Hospital (2017SZ-035). Semen samples were processed using a 40% density gradient of PureSperm (Nidacon International, Molndal, Sweden) through centrifugation at room temperature (500×$g$, 30 min) and washed three times with phosphate-buffered saline (PBS). The obtained spermatozoa were utilized for immunofluorescence staining.

## Antibodies

Rabbit anti-CCDC113 generated from Dia-an Biotech (Wuhan, China) was diluted at 1:500 for western blotting and a 1: 25 dilution for immunofluorescence. Mouse anti-α-TUBULIN antibody (AC012, Abclonal) was used at a 1:5000 dilution for western blotting. Mouse anti-GFP antibody (M20004, Abmart) was used at a 1:2000 dilution for western blotting. Rabbit anti-MYC antibody (BE2011, EASYBIO) was used at a 1:2000 dilution for western blotting. Rabbit anti-FLAG antibody (20543-1 AP, Proteintech) was used at a 1:2000 dilution for western blotting. Mouse anti-ac-TUBULIN antibody (T7451, Sigma-Aldrich) was used at a 1:200 dilution for immunofluorescence. Mouse anti-α/β-TUBULIN antibody (ab44928, Abcam) was used at a 1: 100 dilution for immunofluorescence. Rabbit anti-DRC2 antibody (NBP2-84617, Novus) was used at a 1:100 dilution for immunofluorescence. Rabbit anti-CFAP91 antibody (bs-9823R, Bioss) was used at a 1:100 dilution for immunofluorescence. In-house-generated mouse anti-SUN5 antibody targeting the SUN5 SUN domain (aa193–373) was used at a 1:100 dilution for immunofluorescence analysis. Rat anti-CENTLEIN antibody, generated by Absea Biotechnology Ltd (Beijing, China), was diluted at a 1:20 dilution for immunofluorescence. Rabbit anti-SPATA6 antibody (11849-1 AP, Proteintech) was used at a 1:100 dilution for immunofluorescence. The Alexa Fluor 488 conjugate of lectin PNA (1:400, L21409, Thermo Fisher) was used for immunofluorescence. The secondary antibodies were goat anti-rabbit FITC (1:200, ZF-0311, Zhong Shan Jin Qiao), goat anti-rabbit TRITC (1:200, ZF-0316, Zhong Shan Jin Qiao), goat anti-mouse FITC (1:200, ZF-0312, Zhong Shan Jin Qiao), and goat anti-mouse TRITC (1:200, ZF0313, Zhong Shan Jin Qiao).

## Sperm motility and sperm count assays

The cauda epididymis was isolated from the male mice of the different genotypes. Sperm were released in PBS (Gibco, C14190500BT) from the incisions of the cauda epididymis for 10 min at 37°C.

And then the swim-up suspension was used for the analysis of sperm motility with a microscope through a ×20 phase objective. Viewing areas in each chamber were imaged using a CCD camera. The samples were analyzed via computer-assisted semen analysis (CASA) using the Minitube Sperm Vision Digital Semen Evaluation System (12500/1300, Minitube Group, Tiefenbach, Germany) and were also analyzed by CASA. The incubated sperm number was counted with a hemocytometer.

## Histology staining
As previously reported (*Wang et al., 2018*), the testes and cauda epididymis were dissected after euthanasia, and fixed with Bouin's fixative for 24 hr at 4°C, then the testes were dehydrated with graded ethanol and embedded in paraffin. For histological analysis, the 5 µm sections were cut and covered on glass slides. Sections were stained with H&E and PAS for histological analysis after deparaffinization.

## Electron microscopy analysis
The cauda epididymis and testis were dissected and fixed in 2.5% (vol/vol) glutaraldehyde in 0.1 M cacodylate buffer at 4°C overnight. After washing in 0.1 M cacodylate buffer, samples were cut into small pieces of approximately 1 mm³, then immersed in 1% OsO4 for 1 hr at 4°C. Samples were dehydrated through a graded acetone series (50%, 60%, 70%, 80%, 90%, 95%, 100%) and embedded in resin (DDSA, NMA, enhancer, 812) for staining. Ultrathin sections were cut and stained with uranyl acetate and lead citrate. Images were acquired and analyzed using a JEM-1400 TEM.

## Scanning electron microscopy
The trachea was fixed in 2.5% glutaraldehyde solution overnight, and dehydrated in graded ethanol, subjected to drying, and coated with gold. The images were acquired and analyzed using SU8010 scanning electron microscope.

## Immunofluorescence staining
The testis albuginea was peeled and incubated with collagenase IV and hyaluronidase in PBS for 15 min at 37°C, then washed twice with PBS. Next, fixed with 4% PFA for 5 min, and then coated on slide glass to dry out. The slides were washed with PBS three times and then treated with 0.5% Triton X-100 for 5 min, and blocked with 5% BSA for 30 min. Added the primary antibodies and incubated at 4°C overnight, followed by incubating with a second antibody and DAPI. The images were taken using LSM880 and SP8 microscopes.

The mouse testis was immediately dissected and fixed with 2% paraformaldehyde in 0.05% PBST (PBS with 0.05% Triton X-100) at room temperature for 5 min. The fixed sample was placed on a slide glass and squashed by placing a coverslip on top and pressing down. The sample was immediately flash-frozen in liquid nitrogen, and the slides were stored at −80°C for further immunofluorescence experiments. After removing the coverslips, the slides were washed with PBS three times and then treated with 0.1% Triton X-100 for 10 min, rinsed three times in PBS, and blocked with 5% bovine serum albumin (Amresco, AP0027). The primary antibody was added to the sections and incubated at 4°C overnight, followed by incubation with the secondary antibody. The nuclei were stained with DAPI. The immunofluorescence images were taken immediately using an LSM 780 microscope (Zeiss) or SP8 microscope (Leica).

Spermatozoa were released from the cauda epididymis in PBS at 37°C for 15 min, then were spread on glass slides for morphological observation or immunostaining. After air drying, spermatozoa were fixed in 4% PFA for 5 min at room temperature, and slides were washed with PBS three times and blocked with 5% BSA for 30 min at room temperature. The primary antibodies were added to the sections and incubated at 4°C overnight, followed by incubation with the secondary antibody. The nuclei were stained with DAPI and images were taken using an LSM 880 microscope (Zeiss) or SP8 microscope (Leica).

## Immunoprecipitation
Transfected cells were lysed in a lysis buffer (50 mM HEPES, pH 7.4, 250 mM NaCl, 0.1% NP-40 containing PIC and PMSF) on ice for 30 min and centrifuged at 12,000 rpm at 4°C for 15 min, cell lysates were incubated with primary antibody about 12 hr at 4°C and then incubated with protein

A-Sepharose (GE, 17-1279-03) for 3 hr at 4°C, then washed three times with lysed buffer and subjected to immunoblotting analysis.

## Statistical analysis

All the experiments were repeated at least three times, and the results are presented as the mean ± SD. The statistical significance of the differences between the mean values for the different genotypes was measured by the Student's t-test with a paired, two-tailed distribution. The data were considered significant when the p-value was less than 0.05(*), 0.01(**), 0.001(***), or 0.0001(****).

## Acknowledgements

This work was supported by the National Science Fund for Distinguished Young Scholars (Grant No. 81925015), National Natural Science Foundation of China (Grant No. 32230029, 82371615), National Key Research and Development Program of China (Grant No. 2022YFC2702600), and Science and Technology Project of Guangzhou (Grant No. 2023A03J0886, 2023A03J0871).

## Additional information

### Funding

| Funder | Grant reference number | Author |
| --- | --- | --- |
| National Science Fund for Distinguished Young Scholars | 81925015 | Wei Li |
| National Natural Science Foundation of China | 32230029 | Wei Li |
| National Key Research and Development Program of China | 2022YFC2702600 | Wei Li |
| Science and Technology Project of Guangzhou | 2023A03J0886 | Wei Li |
| Science and Technology Project of Guangzhou | 2023A03J0871 | Chao Liu |
| National Natural Science Foundation of China | 82371615 | Yuzhuo Yang |

The funders had no role in study design, data collection and interpretation, or the decision to submit the work for publication.

### Author contributions

Bingbing Wu, Conceptualization, Resources, Data curation, Software, Formal analysis, Validation, Investigation, Visualization, Methodology, Writing – original draft, Project administration, Writing – review and editing; Chenghong Long, Conceptualization, Data curation, Formal analysis, Validation, Investigation, Visualization, Methodology, Writing – original draft, Writing – review and editing; Yuzhuo Yang, Zhe Zhang, Resources, Investigation; Shuang Ma, Yanjie Ma, Huafang Wei, Jinghe Li, Hui Jiang, Investigation; Wei Li, Conceptualization, Resources, Supervision, Funding acquisition, Methodology, Project administration, Writing – review and editing; Chao Liu, Conceptualization, Resources, Supervision, Funding acquisition, Validation, Project administration, Writing – review and editing

### Author ORCIDs

Wei Li ⓘ https://orcid.org/0000-0002-6235-0749
Chao Liu ⓘ https://orcid.org/0000-0002-8844-0697

### Ethics

The sperm donation candidates in this study were healthy young Chinese men. Each participant underwent a thorough medical examination and completed an extensive medical and social questionnaire to ensure the exclusion of individuals with genetic or significant medical issues, as outlined in

the Basic Standard and Technical Norms of Human Sperm Bank published by the Chinese Ministry of Health. Individuals who smoked, abused drugs, or were heavy drinkers were also excluded from the study. Those who remained eligible signed informed consent forms for voluntary sperm donation and agreed to reside in Beijing for a minimum of six months. The sperm bank documented each participant's age, date of birth, and date of semen collection. Ethical approval for this study was granted by the Reproductive Study Ethics Committee of Peking University Third Hospital (2017SZ-035).

All the animal experiments were performed according to approved institutional animal care and use committee (IACUC) protocols (# 08-133) of the Institute of Zoology, Chinese Academy of Sciences.

Reviewer #1 (Public review): https://doi.org/10.7554/eLife.98016.3.sa1
Reviewer #2 (Public review): https://doi.org/10.7554/eLife.98016.3.sa2
Author response https://doi.org/10.7554/eLife.98016.3.sa3

### Data availability

All data generated or analysed during this study are included in the manuscript and supporting files; source data files have been provided for *Figures 1, 2, 4 and 7*, *Figure 1—figure supplement 1*, *Figure 2—figure supplement 1* and *Figure 4—figure supplement 1*.

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
