## [Editor Report · eLife Assessment]

This study presents an **important** finding on sperm flagellum and HTCA stabilization. The evidence supporting the authors' claims is **convincing**. The work will be of broad interest to cell and reproductive biologists working on cilium and sperm biology.

---

## [Referee Report · Reviewer #1 (Public review)]

In this paper, Wu et al. investigated the physiological roles of CCDC113 in sperm flagellum and HTCA stabilization by using CRISPR/Cas knockouts mouse models, co-IP and single sperm imaging. They find that CCDC113 localizes in the linker region among radial spokes, the nexin-dynein regulatory complex (N-DRC), and doublet microtubules (DMTs) RS, N-DRC and DMTs and interacts with axoneme-associated proteins CFAP57 and CFAP91, acting as an adaptor protein that facilitates the linkage between RS, N-DRC and DMTs within the sperm axoneme. They show the disruption of CCDC113 produced spermatozoa with disorganized sperm flagella and CFAP91, DRC2 could not colocalize with DMTs in *Ccdc113*^*–/–*^ spermatozoa. Interestingly, the data also indicate that CCDC113 could localize on the HTCA region, and interact with HTCA-associated proteins. The knockout of Ccdc113 could also produce acephalic spermatozoa. By using Sun5 and Centlein knockout mouse models, the authors further find SUN5 and CENTLEIN are indispensable for the docking of CCDC113 to the implantation site on the sperm head. Overall, the experiments were designed properly and performed well to support the authors' observation in each part. Furthermore, the study's findings offer valuable insights into the physiological and developmental roles of CCDC113 in the male germ line, which can provide insight into impaired sperm development and male infertility. The conclusions of this paper are mostly well supported by data, but some points need to be clarified and discussed.

(1) In Fig. 1, a sperm flagellum protein, which is far way from CCDC113, should be selected as a negative control to exclude artificial effects in co-IP experiments.

(2) Whether the detachment of sperm head and tail in *Ccdc113*^*–/–*^ mice is a secondary effect of the sperm flagellum defects? The author should discuss this point.

(3) Given that some cytoplasm materials could be observed in *Ccdc113*^*–/–*^ spermatozoa (Fig. 5A), whether CCDC113 is also essential for cytoplasmic removal?

(4) Although CCDC113 could not bind to PMFBP1, the localization of CCDC113 in *Pmfbp1*^*–/–*^ spermatozoa should be also detected to clarify the relationship between CCDC113 and SUN5-CENTLEIN-PMFBP1.

Comments on revisions:

The authors addressed all my concerns. The manuscript was greatly improved.

---

## [Referee Report · Reviewer #2 (Public review)]

Summary:

In the present study, the authors select the coiled-coil protein CCDC113 and revealed its expression in the stages of spermatogenesis in the testis as well as in the different steps of spermiogenesis with expression also mapped in the different parts of the epididymis. Gene deletion led to male infertility in CRISPR-Cas9 KO mice and PAS staining showed defects mapped in the different stages of the seminiferous cycle and through the different steps of spermiogenesis. EM and IF with several markers of testis germ cells and spermatozoa in the epididymis indicated defects in flagella and head-to-tail coupling for flagella as well as acephaly. The authors' co-IP experiments of expressed CCDC113 in HEK293T cells indicated an association with CFAP91 and DRC2 as well as SUN5 and CENTLEIN.

The authors propose that CCDC113 connects CFAP91 and DRC2 to doublet microtubules of the axoneme and CCDC113's association with SUN5 and CENTLEIN to stabilize the sperm flagellum head-to-tail coupling apparatus. Extensive experiments mapping CCDC13 during postnatal development are reported as well as negative co-IP experiments and studies with SUN5 KO mice as well as CENTLEIN KO mice.

Strengths:

The authors provide compelling observations to indicate the relevance of CCDC113 to flagellum formation with potential protein partners. The data are relevant to sperm flagella formation and its coupling to the sperm head.

Weaknesses:

The authors' observations are consistent with the model proposed but the authors' conclusions for the mechanism may require direct demonstration in sperm flagella. The Walton et al paper shows human CCDC96/113 in cilia of human respiratory epithelia. An application of such methodology to the proteins indicated by Wu et al for the sperm axoneme and head-tail coupling apparatus is eagerly awaited as a follow-up study.

---

## [Author Response]

The following is the authors’ response to the original reviews.

This study presents a valuable finding on sperm flagellum and HTCA stabilization. The evidence supporting the authors' claims is incomplete. The work will be of broad interest to cell and reproductive biologists working on cilium and sperm biology.

We thank the Editor and the two reviewers for their time and thorough evaluation of our manuscript. We greatly appreciate their valuable guidance on improving our study. In the revised manuscript, we have conducted additional experiments and provided quantitative data in response to the reviewers' comments. Furthermore, we have refined the manuscript and added further context to elucidate the significance of our findings for the readers.

**Public Reviews:**

**Reviewer #1 (Public Review):**
In this paper, Wu et al. investigated the physiological roles of CCDC113 in sperm flagellum and HTCA stabilization by using CRISPR/Cas knockouts mouse models, co-IP, and single sperm imaging. They find that CCDC113 localizes in the linker region among radial spokes, the nexin-dynein regulatory complex (N-DRC), and doublet microtubules (DMTs) RS, N-DRC, and DMTs and interacts with axoneme-associated proteins CFAP57 and CFAP91, acting as an adaptor protein that facilitates the linkage between RS, N-DRC, and DMTs within the sperm axoneme. They show the disruption of CCDC113 produced spermatozoa with disorganized sperm flagella and CFAP91, DRC2 could not colocalize with DMTs in *Ccdc113*^*-/-*^ spermatozoa. Interestingly, the data also indicate that CCDC113 could localize on the HTCA region, and interact with HTCA-associated proteins. The knockout of Ccdc113 could also produce acephalic spermatozoa. By using Sun5 and Centlein knockout mouse models, the authors further find SUN5 and CENTLEIN are indispensable for the docking of CCDC113 to the implantation site on the sperm head. Overall, the experiments were designed properly and performed well to support the authors' observation in each part. Furthermore, the study's findings offer valuable insights into the physiological and developmental roles of CCDC113 in the male germ line, which can provide insight into impaired sperm development and male infertility. The conclusions of this paper are mostly well supported by data, but some points need to be clarified and discussed.

We thank Reviewer #1 for his or her critical reading and the positive assessment.

(1) In Figure 1, a sperm flagellum protein, which is far away from CCDC113, should be selected as a negative control to exclude artificial effects in co-IP experiments.

We greatly appreciate Reviewer #1’s insightful suggestion. In response, we selected two sperm outer dense fiber proteins, ODF1 and ODF2, which are located distant from the sperm axoneme, as negative controls in the co-IP experiments. As shown in Figure 1- figure supplement 1A and B, neither ODF1 nor ODF2 bound to CCDC113, indicating the interaction observed in Figure 1 is not an artifact.

(2) Whether the detachment of sperm head and tail in *Ccdc113*^*–/–*^ mice is a secondary effect of the sperm flagellum defects? The author should discuss this point.

Good question. Considering that CCDC113 is localized in the sperm neck region and interacts with SUN5 and CENTLEIN, it may play a direct role in connecting the sperm head and tail. Indeed, PAS staining revealed that *Ccdc113*^*–/–*^ sperm heads exhibit abnormal orientation in stages V–VIII of the seminiferous epithelia (Figure 6C-D). Furthermore, transmission electron microscopy (TEM) analysis indicated that the absence of CCDC113 caused detachment of the damaged coupling apparatus from the sperm head in step 9–11 spermatids (Figure 6E). These results suggest that the detachment of the sperm head and tail in *Ccdc113*^*–/–*^ mice may not be a secondary effect of sperm flagellum defects. We have discussed this point further below:

“CCDC113 can interact with SUN5 and CENTLEIN, but not PMFBP1 (Figure 7A-C), and left on the tip of the decapitated tail in *Sun5*^*–/–*^ and *Centlein*^*–/–*^ spermatozoa (Figure 7K and L). Furthermore, CCDC113 colocalizes with SUN5 in the HTCA region, and immunofluorescence staining in spermatozoa shows that SUN5 is positioned closer to the sperm nucleus than CCDC113 (Figure 7G). Therefore, SUN5 and CENTLEIN may be closer to the sperm nucleus than CCDC113. PAS staining revealed that *Ccdc113*^*–/–*^ sperm heads are abnormally oriented in stages V–VIII seminiferous epithelia (Figure6 C and D), and TEM analysis further demonstrated that the disruption of CCDC113 causes the detachment of the destroyed coupling apparatus from the sperm head in step 9–11 spermatids (Figure 6E). All these results suggest that the detachment of sperm head and tail in *Ccdc113*^*–/–*^ mice may not be a secondary effect of sperm flagellum defects.”

(3) Given that some cytoplasm materials could be observed in *Ccdc113*^–/–^ spermatozoa (Fig. 5A), whether CCDC113 is also essential for cytoplasmic removal?

Good question. Unremoved cytoplasm could be detected in spermatozoa by using transmission electron microscopy (TEM) analysis, including disrupted mitochondria, damaged axonemes, and large vacuoles. These observations indicate defects in cytoplasmic removal in *Ccdc113*^*–/–*^ mice. We have discussed this point as below:

“Moreover, TEM analysis detected excess residual cytoplasm in spermatozoa, including disrupted mitochondria, damaged axonemes, and large vacuoles, indicating defects in cytoplasmic removal in *Ccdc113*^*–/–*^ mice (Figure 5A).”

(4) Although CCDC113 could not bind to PMFBP1, the localization of CCDC113 in *Pmfbp1*^*–/–*^ spermatozoa should be also detected to clarify the relationship between CCDC113 and SUN5-CENTLEIN-PMFBP1.

We appreciate Reviewer #1’s suggestion. We have analyzed the localization of CCDC113 in *Pmfbp1*^*–/–*^ spermatozoa and found that CCDC113 was located at the tip of the decapitated tail in *Pmfbp1*^*–/–*^ spermatozoa (Figure 7K and L). This finding has been incorporated into the revised manuscript as below:

“To further elucidate the functional relationships among CCDC113, SUN5, CENTLEIN, and PMFBP1 at the sperm HTCA, we examined the localization of CCDC113 in *Sun5*^*–/–*^, *Centlein*^*–/–*^, and *Pmfbp1*^*–/–*^ spermatozoa. Compared to the control group, CCDC113 was predominantly localized on the decapitated flagellum in *Sun5*^*–/–*^, *Centlein^–/–^*, and *Pmfbp1^–/–^* spermatozoa (Figure 7K and L), indicating SUN5, CENTLEIN, and PMFBP1 are crucial for the proper docking of CCDC113 to the implantation site on the sperm head. Taken together, these data demonstrate that CCDC113 cooperates with SUN5 and CENTLEIN to stabilize the sperm HTCA and anchor the sperm head to the tail.”

**Reviewer #2 (Public Review):**
Summary:In the present study, the authors select the coiled-coil protein CCDC113 and revealed its expression in the stages of spermatogenesis in the testis as well as in the different steps of spermiogenesis with expression also mapped in the different parts of the epididymis. Gene deletion led to male infertility in CRISPR-Cas9 KO mice and PAS staining showed defects mapped in the different stages of the seminiferous cycle and through the different steps of spermiogenesis. EM and IF with several markers of testis germ cells and spermatozoa in the epididymis indicated defects in flagella and head-to-tail coupling for flagella as well as acephaly. The authors' co-IP experiments of expressed CCDC113 in HEK293T cells indicated an association with CFAP91 and DRC2 as well as SUN5 and CENTLEIN.The authors propose that CCDC113 connects CFAP91 and DRC2 to doublet microtubules of the axoneme and CCDC113's association with SUN5 and CENTLEIN to stabilize the sperm flagellum head-to-tail coupling apparatus. Extensive experiments mapping CCDC13 during postnatal development are reported as well as negative co-IP experiments and studies with SUN5 KO mice as well as CENTLEIN KO mice.Strengths:The authors provide compelling observations to indicate the relevance of CCDC113 to flagellum formation with potential protein partners. The data are relevant to sperm flagella formation and its coupling to the sperm head.

We are grateful to Reviewer #2 for his or her recognition of the strength of this study.

Weaknesses:The authors' observations are consistent with the model proposed but the authors' conclusions for the mechanism may require direct demonstration in sperm flagella. The Walton et al paper shows human CCDC96/113 in cilia of human respiratory epithelia. An application of such methodology to the proteins indicated by Wu et al for the sperm axoneme and head-tail coupling apparatus is eagerly awaited as a follow-up study.

We thank Reviewer 2 for his/her kindly help in improving the manuscript. We now understand that directly detection of CCDC113 precise localization in sperm axoneme and head-tail coupling apparatus (HTCA) using cryo-electron microscopy (cryo-EM) could powerfully strengthen our model. Recent advances in cryo-EM have indeed advanced our understanding of axonemal structures analysis of axonemal structures and determined the structures of native axonemal DMTs from mouse, bovine, and human sperm (Leung et al., 2023; Zhou et al., 2023). However, high-resolution structures of sperm axoneme and HTCA regions, including those involving CCDC113, have yet to be fully characterized. Thus, we would like to discuss this point and consider it a valuable direction for future research.

“Given that the cryo-EM of sperm axoneme and HTCA could powerfully strengthen the role of CCDC113 in stabilizing sperm axoneme and head-tail coupling apparatus, it a valuable direction for future research.”

References:

Bazan, R., Schröfel, A., Joachimiak, E., Poprzeczko, M., Pigino, G., & Wloga, D. (2021). Ccdc113/Ccdc96 complex, a novel regulator of ciliary beating that connects radial spoke 3 to dynein g and the nexin link. PLoS Genet, 17(3), e1009388.

Ghanaeian, A., Majhi, S., McCafferty, C. L., Nami, B., Black, C. S., Yang, S. K., Legal, T., Papoulas, O., Janowska, M., Valente-Paterno, M., Marcotte, E. M., Wloga, D., & Bui, K. H. (2023). Integrated modeling of the Nexin-dynein regulatory complex reveals its regulatory mechanism. Nat Commun, 14(1), 5741.

Leung, M. R., Zeng, J., Wang, X., Roelofs, M. C., Huang, W., Zenezini Chiozzi, R., Hevler, J. F., Heck, A. J. R., Dutcher, S. K., Brown, A., Zhang, R., & Zeev-Ben-Mordehai, T. (2023). Structural specializations of the sperm tail. Cell, 186(13), 2880-2896.e2817

Walton, T., Gui, M., Velkova, S., Fassad, M. R., Hirst, R. A., Haarman, E., O'Callaghan, C., Bottier, M., Burgoyne, T., Mitchison, H. M., & Brown, A. (2023). Axonemal structures reveal mechanoregulatory and disease mechanisms. Nature, 618(7965), 625-633.

Zhou, L., Liu, H., Liu, S., Yang, X., Dong, Y., Pan, Y., Xiao, Z., Zheng, B., Sun, Y., Huang, P., Zhang, X., Hu, J., Sun, R., Feng, S., Zhu, Y., Liu, M., Gui, M., & Wu, J. (2023). Structures of sperm flagellar doublet microtubules expand the genetic spectrum of male infertility. Cell, 186(13), 2897-2910.e2819.

**Recommendations for the authors:**

**Reviewer #1 (Recommendations For The Authors):**
(1) Please provide full gel for the Figure 2C experiment (could be as a supplementary file).

Thanks for your insightful suggestions. We have replaced Figure 2C and provided the full gel in Figure 2-figure supplement 1A.

(2) The authors write on Line 163 "In contrast, the flagellum staining appeared reduced in *Ccdc113*^*–/–*^ seminiferous tubules (Fig. 2J, red asterisk)." However, the magnification of the pictures is not sufficient to distinguish anything in the panel mentioned, please provide others.

Many thanks for pointing this out. We have provided the iconic figure to show the flagella defect in seminiferous tubules.

(3) Please add statistical p-values for figures.

Thanks for your valuable advice. We have added statistical p-values to the figures in the revised manuscript.

(4) Line 128: Should "speculate" be "speculated"?

Thank you for pointing out this problem. We have corrected it in the revised manuscript, as shown below:

“Given that CFAP91 has been reported to stabilize RS on the DMTs (Bicka et al., 2022; Dymek et al., 2011; Gui et al., 2021) and cryo-EM analysis shows that CCDC113 is closed to DMTs, we speculated that CCDC113 may connect RS to DMTs by binding to CFAP91 and microtubules.”

(5) In lines 384-385, more "-" is typed.

Thank you for pointing out this problem. We have corrected it in the revised manuscript, as shown below:

“Furthermore, CCDC113 colocalizes with SUN5 in the HTCA region, and immunofluorescence staining in spermatozoa shows that SUN5 is closer to the sperm nucleus than CCDC113 (Figure 7G). Therefore, SUN5 and CENTLEIN may be closer to the sperm nucleus than CCDC113.”

(6) In general, the article has many typos and should be professionally proofread.

Many thanks for pointing this out. We have thoroughly revised the manuscript with the assistance professional proofreading.

**Reviewer #2 (Recommendations For The Authors):**
Can the authors indicate in the Materials and Methods if n=3 biological replicates were done for all co-IP, EM, LM, and IF studies? The statistical analysis section indicates this but quantification is missing for most figures including co-IP, most IF, PAS staining, EM, etc.

We thank Reviewer 2 for the insightful comments and guidance to improve our data quality. All the experiments in this study were repeated at least three times to ensure reproducibility. We have quantified the co-IP experiments in Figures 1C-H and 7A-F, the IF data in Figures 2K, 5C, and 5D, as well as the PAS staining in Figure 6C. Since electron microscopy samples require very little testicular tissue and the sections obtained are very thin, the likelihood of capturing sections specifically at the sperm head-tail junction is considerably low. This challenge makes it difficult to perform quantitative analysis and statistical evaluation in the TEM experiment. To address this limitation, we have quantified the percentage of *Ccdc113*^*–/–*^ sperm heads with abnormal orientation in stages V–VIII of the seminiferous epithelium to indicate impaired head-to-tail anchorage.

Figure S2 is compelling and might be indicated as a major figure instead of a supplementary figure.

We appreciate the positive comment. We have included it as a major figure in Figure 3F.

Figure 4A may be incomplete. Data sets for RNA expression suggest high expression in the ovary and other organs in males and females including the brain and are not indicated by the authors. Figure 4A may be considered for removal with a more complete study for another paper.

Thank you for pointing out this issue. We reviewed RNA expression data from various tissues using RNA-Seq data from Mouse ENCODE (https://www.ncbi.nlm.nih.gov/gene/244608) and found that CCDC113 is highly expressed in the testis, but not significantly in the ovary and brain (Figure 4- figure supplement 1A). Additionally, we re-evaluated CCDC113 protein levels in the spleen, lung, kidney, testis, intestine, stomach, brain, and ovary, confirming that it is highly expressed in the testes, with negligible expression in the ovary and brain (Figure 4- figure supplement 1B). In line with Reviewer 2's suggestion, we have removed Figure 4A in the revised manuscript.

There are grammatical errors throughout the manuscript and Figure 7 is truncated.

Thank you for pointing out this problem. We have thoroughly revised the manuscript with the assistance professional proofreading.

The Introduction and Discussion parts of the paper may need some clarification for the general reader. The material in the "Additional Context " section of the critique below may be a helpful place to introduce what a stage is, and the steps in germ cell development in the testis with the latter of course where and when the flagellum develops.

We appreciate your valuable suggestions. We have referred to the material in the “Additional Context” section to introduce the stages of spermatogenesis and the steps in germ cell development in the testis in the introduction and results.

“Male fertility relies on the continuous production of spermatozoa through a complex developmental process known as spermatogenesis. Spermatogenesis involves three primary stages: spermatogonia mitosis, spermatocyte meiosis, and spermiogenesis. During spermiogenesis, spermatids undergo complex differentiation processes to develop into spermatozoa, which includes nuclear elongation, chromatin remodeling, acrosome formation, cytoplasm elimination, and flagellum development (Hermo et al., 2010).”

Hermo, L., Pelletier, R. M., Cyr, D. G., & Smith, C. E. (2010). Surfing the wave, cycle, life history, and genes/proteins expressed by testicular germ cells. Part 1: background to spermatogenesis, spermatogonia, and spermatocytes. Microscopy research and technique, 73(4), 241–278. https://doi.org/10.1002/jemt.20783

“Pioneering work in the mid-1950s used the PAS stain in histologic sections of mouse testis to visualize glycoproteins of the acrosome and Golgi in seminiferous tubules (Oakberg, 1956). The pioneers discovered in cross-sectioned seminiferous tubules the association of differentiating germ cells with successive layers to define different stages that in mice are twelve, indicated as Roman numerals (XII). For each stage, different associations of maturing germ cells were always the same with early cells in differentiation at the periphery and more mature cells near the lumen. In this way, progressive differentiation from stem cells to mitotic, meiotic, acrosome-forming, and post-acrosome maturing spermatocytes was mapped to define spermatogenesis with the XII stages in mice representing the seminiferous cycle. The maturation process from acrosome-forming cells to mature spermatocytes is defined as spermiogenesis with 16 different steps that are morphologically distinct spermatids (O'Donnell L, 2015).”

Oakberg, E. F. (1956). A description of spermiogenesis in the mouse and its use in analysis of the cycle of the seminiferous epithelium and germ cell renewal. The American journal of anatomy, 99(3), 391-413. https://doi.org/10.1002/aja.1000990303

O'Donnell L. (2015). Mechanisms of spermiogenesis and spermiation and how they are disturbed. Spermatogenesis, 4(2), e979623. https://doi.org/10.4161/21565562.2014.979623

For the Discussion, the authors indicate that the function of CCDC113 in mammals is unknown yet the authors point to the work of Walton et al on human respiratory epithelia that points to a function for CCDC96/113. The work in the manuscript here does indicate a role in sperm flagella and the head-to-tail coupling apparatus but remains descriptive until the methodology of Walton et al is applied. Hopefully, the authors will consider it for a follow-up study.

Thank you for pointing out this problem. We have revised this part and highlighted the Walton et al’s work in the Discussion.

“CCDC113 is a highly evolutionarily conserved component of motile cilia/ﬂagella. Studies in the model organism, *Tetrahymena thermophila*, have revealed that CCDC113 connects RS3 to dynein g and the N-DRC, which plays essential role in cilia motility (Bazan et al., 2021; Ghanaeian et al., 2023). Recent studies have also identified the localization of CCDC113 within the 96-nm repeat structure of the human respiratory epithelial axoneme, and localizes to the linker region among RS, N-DRC and DMTs (Walton et al., 2023). In this study, we reveal that CCDC113 is indispensable for male fertility, as Ccdc113 knockout mice produce spermatozoa with flagellar defects and head-tail linkage detachment (Figure 3D).”

“Overall, we identified CCDC113 as a structural component of both the flagellar axoneme and the HTCA, where it performs dual roles in stabilizing the sperm axonemal structure and maintaining the structural integrity of HTCA. Given that the cryo-EM of sperm axoneme and HTCA could powerfully strengthen the role of CCDC113 in stabilizing sperm axoneme and head-tail coupling apparatus, it a valuable direction for future research.”

The Discussion may be focused on the key aspects of CCDC113 related to sperm flagella and the head-to-tail coupling apparatus that represent a genuine advance. The more speculative parts of the Discussion that have not been addressed by experimentation in the Results section may be considered for removal in the Discussion section.

Thank you for pointing out this. We have removed the speculative parts of the Discussion that have not been addressed by experimentation in the Results section.

Additional Context to help readers understand the significance of the work:Pioneering work in the mid-1950s used the periodic acid Schiff (PAS) stain in histologic sections of rodent testis to visualize glycoproteins of the acrosome and Golgi in seminiferous tubules. The pioneers discovered in cross-sectioned seminiferous tubules the association of differentiating germ cells with successive layers to define different stages that in mice are twelve, indicated as Roman numerals (XII). For each stage, different associations of maturing germ cells were always the same with early cells in differentiation at the periphery and more mature cells near the lumen. In this way, progressive differentiation from stem cells to mitotic, meiotic, acrosome-forming, and post-acrosome maturing spermatocytes was mapped to define spermatogenesis with the XII stages in mice representing the seminiferous cycle. The maturation process from acrosome-forming cells to mature spermatocytes is defined as spermiogenesis with 19 different steps that are morphologically distinct spermatids. It is from steps 8-19 of spermiogenesis that the formation of the flagellum takes place. Final maturation occurs in the epididymis as sperm move through the caput, corpus, and cauda of the organ with motile spermatozoa generated.

Thank you very much!